# No obligatory trade-off between the use of space and time for working memory

Eelke de Vries [1]✉, George Fejer [2] & Freek van Ede [1]✉

Space and time can each act as scaffolds for the individuation and selection of visual objects in working memory. Here we ask whether there is a trade-off between the use of space and time for visual working memory: whether observers will rely less on space, when memoranda can additionally be individuated through time. We tracked the use of space through directional biases in microsaccades after attention was directed to memory contents that had been encoded simultaneously or sequentially to the left and right of fixation. We found that spatial gaze biases were preserved when participants could (Experiment 1) and even when they had to (Experiment 2) additionally rely on time for object individuation. Thus, space remains a profound organizing medium for working memory even when other organizing sources are available and utilized, with no evidence for an obligatory trade-off between the use of space and time.

[1] Department of Experimental and Applied Psychology, Institute for Brain and Behavior Amsterdam, Vrije Universiteit Amsterdam, Amsterdam, The Netherlands. [2] Department of Psychology, Cognitive Psychology, University of Konstanz, Konstanz, Germany. ✉email: e3.de.vries@vu.nl; freek.van.ede@vu.nl

Our perception of the visual world around us is structured by the space and time in which events unfold. Consequently, space and time may also be relevant for visual inputs that are no longer visible but instead are held available in working memory for upcoming use. How does the brain represent the spatial locations and temporal arrangements of recently seen events, and utilize these to individuate objects and enable later object selection from working memory?

Ample research has demonstrated a special 'scaffolding' role of space for visual working memory—even when space itself is not the target memory feature. Space may play a role in binding the features that belong to the same object, thereby supporting object individuation[1–6]. Space may further be used as a medium for subsequent attentional selection and prioritization of specific objects within working memory[7–12]. The use of space is further consistent with sensory-recruitment models of working memory[13–17] whereby retinotopically organized visual cortex is also utilized for visual retention in working memory—thus 'recycling' existing visual-spatial coding architecture in the human brain[18].

At the same time, it has recently become clear that time can also play an important role in working memory. Like space, time (such as temporal order at encoding) may also enable object individuation[2,19–24] and object selection[8,13,25,26] in working memory. Moreover, like for space, time may serve working memory 'incidentally', i.e., even when it is not the target memory feature[27,28]. Accordingly, when the brain can use time to individuate and select objects from working memory, it is conceivable that there may be less need to (also) rely on space to serve the same purpose of individuation and subsequent attentional selection and prioritization.

Here we examine whether there is a trade-off between the incidental use of space and time for visual working memory. In other words, will observers rely less on space, when objects can additionally be individuated through time? We consider three possible scenarios. The first possibility (Fig. 1a) is that there is an automatic reliance on space as a scaffold for working memory, even when the brain can or must also rely on time. A second possibility (Fig. 1b) is that the availability of time for object individuation decreases but does not abolish the complementary reliance on space. A third possibility (Fig. 1c) is that the availability of time for object individuation eliminates the incidental use of space altogether.

To address this question, we manipulated whether or not participants could (Experiment 1) or had to (Experiment 2) rely on temporal order to individuate individual objects in working memory, while always also allowing object individuation through space. To track the incidental use of space, we built on a recent finding by van Ede et al.[11] where it was demonstrated that attending to an object in visual working memory, biases gaze (microsaccades) in the direction of its memorized location (as also replicated in refs. [29,30]). This 'gaze bias' provides an ideal approach for our current question because it reflects a direct index of the incidental use of space for working memory (also see ref. [31]), that is observed even when space is not the target memory feature.

In Experiment 1, the spatial separation between two memory objects was always the same, but we varied the temporal separation by presenting objects simultaneously (only spatial separation) or sequentially (spatial and temporal separation). Space is the obvious way to individuate objects when they are presented simultaneously—even when space is technically not needed for the task. However, when objects are encoded sequentially, they can (additionally) be individuated by time (i.e., order of encoding) and space may become less important (see scenarios in Fig. 1). If so, this should be reflected in our spatial gaze signature of object selection from working memory.

Participants in the first experiment could, but did not have to, rely on time for object individuation and selection. In Experiment 2, we therefore, always presented objects sequentially and this time cued relevant objects in working memory either through their colour (as in Experiment 1) or their temporal order (first or second). This allowed us to compare the use of space for mnemonic object selection when participants were forced to use time (temporal order cues) or not (colour cues) to complete the task.

To preview our results, in both experiments, we found that spatial biases in gaze were preserved when participants could (Experiment 1) or had to (Experiment 2) additionally rely on time. This suggests that space remains important for working memory, even when time is also available to serve object individuation and selection. In other words, we report no evidence for an obligatory trade-off in the incidental use of space and time for visual working memory.

## Methods

**Participants**. We conducted two experiments with 25 healthy human volunteers in each. Sample size of 25 was set a-priori based on the use of the same sample size in relevant prior studies that used the same outcome measure that was highly robust in each prior case[11,29,30], and is typically observed in each participant[11,32]. We used the same a-priori determined sample

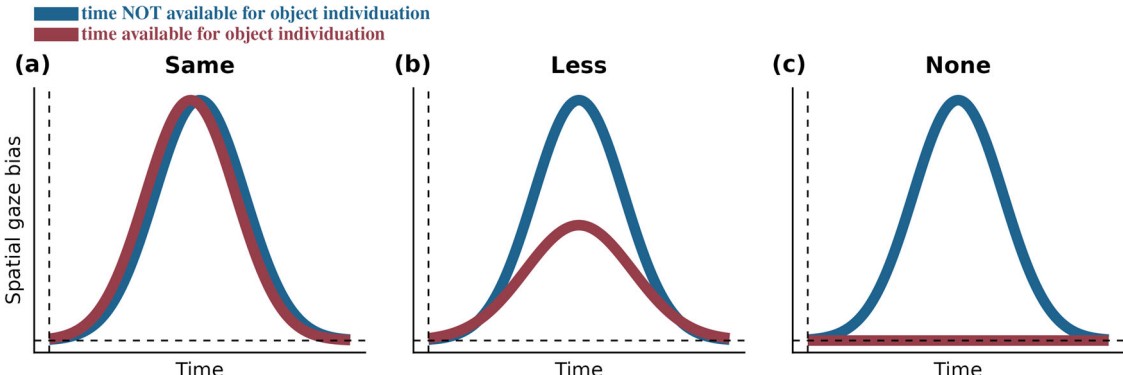

**Fig. 1 Possible scenarios regarding the reliance on space for working memory when time is available or is not available for object individuation.** The incidental use of space is tracked through spatial biases in gaze when a particular object is selected from working memory following a non-spatial colour retrocue (as in ref. [11]). The concurrent availability of time for item individuation and subsequent selection could **a** leave the incidental use of space unaltered, **b** attenuate the incidental use of space, or **c** abolish the incidental use of space for working altogether.

size for both experiments, and report how our key outcome measure replicated between experiments 1 and 2. Through a sensitivity analysis conducted with the 'pwr' package in R[33], we confirmed that our within-subject design, involving 25 participants in each of two conditions per experiment and a significance threshold of $\alpha = 0.05$, provides 80% power to detect an effect size of medium to large magnitude (Cohen's $d = 0.584$). This indicates that our study was sufficiently powered to detect any meaningful effect of a reasonable magnitude, although smaller effects might have remained undetected. Participant recruitment for the two experiments was performed independently. In Experiment 1 (mean age = 27.24 years, age range = 18–39 years, 15 women; 1 left-handed, 1 ambidextrous) two participants were excluded and replaced due to poor eye-tracking quality. In Experiment 2 (mean age = 21.32 years, age range = 18–28 years, 19 women; 2 left-handed) one participant was excluded and replaced due to poor eye-tracking quality. Gender was self-reported. The experiments received approval from the Research Ethics Committee of the Faculty of Behavioural and Movement Sciences at the Vrije Universiteit Amsterdam. Prior to participation, all participants gave their written informed consent, confirmed having normal or corrected-to-normal vision, and were offered course credits or monetary compensation at a rate of €10 per hour. The studies were not preregistered.

**Task and procedure Experiment 1**. Experiment 1 (Fig. 2a) used a 2 (encoding condition: simultaneous, sequential) × 2 (target location: left, right) within-subject design. Participants were instructed to fixate on a central black cross while performing a visual working memory task in which they memorized the orientation of two coloured bars (green, and purple) that were presented on either side of the fixation cross (horizontal displacement of 8 visual degrees). Critically, we varied the temporal separation by presenting the two memory items either simultaneously or sequentially. Time was here thus operationalised as temporal order, which allowed participants to individuate memory objects across time. This manipulation allowed us to investigate whether items that are encoded sequentially (spatial and temporal separation) rely less on space for individuating items in visual working memory than items that were presented simultaneously (only spatial separation).

Stimuli were generated and presented in Presentation (Neurobehavioral Systems) on an LCD monitor (ASUS ROG Strix XG248; 1920 × 1080 pixels; 240 Hz refresh rate) with a solid grey background. Participants were seated in a dimly lit room with their heads resting on a chin rest at a viewing distance of ~75 cm from the display. Each trial began with a variable intertrial interval between 500 and 800 ms during which a central black fixation cross was printed on a grey background. This was followed by two encoding moments (early, late) that each lasted for 250 ms and were separated by an interstimulus interval of 750 ms. 1250 ms after the second encoding moment, the fixation cross briefly changed colour (250 ms) serving as a retrocue to indicate which item would be probed at the end of the trial. The retrocue was 100% valid and was always followed by a second memory interval (1250 ms) after which the fixation cross turned grey, indicating that the participant could initiate their response by moving the computer mouse to reproduce the memorized orientation of the cued item on a response dial that was always presented centrally around the fixation cross.

Experiment 1 included three separate types of simultaneous trials (both only early; both only late; both early and late), and two separate types of sequential trials (first left then right; first right then left). The timing of early and late encoding moments was fixed across trials, even if either potential encoding moment

contained no items (i.e. when both items were presented simultaneously in either the first or the second potential encoding frame). We collapsed across these sub-types to focus on the main comparison between sequential and simultaneous trials. The five trial types occurred equally often in the experiment and contained an equal number of trials in which we probed the item that happened to be presented on the left or the right side of fixation at encoding.

The colour (green, purple) and position (left, right of fixation) of the stimuli were varied semi-randomly, so that these features were distributed equally for targets and non-targets. Experimental sessions comprised 10 blocks, with 50 trials each. Each block ended with a custom calibration procedure, in which participants were asked to track a black dot with their eyes, while it moved across 7 positions on the screen, in random order. Before beginning the experiment, participants practiced the task for 5–10 min. Experimental sessions lasted ~70 min per participant.

**Task and procedure, Experiment 2**. Experiment 2 (Fig. 3a) followed the same overall set-up as Experiment 1 with several key differences. The critical manipulation in Experiment 2 was that items were either cued through their colour (as in Experiment 1) or through their temporal order (first or second). This manipulation allowed us to investigate whether the incidental use of space is dependent on whether or not participants were forced to use time (temporal order cues) or were not forced to use time (colour cues) to complete the task. In Experiment 2, the two coloured bars were red and blue (however, for illustrative purposes, we adhered to the same colours for Experiments 1 and 2 in our figures, to facilitate a consistent, easy-to-follow visualization). Moreover, the two bars were always presented sequentially. To avoid differences in cue appearance between colour and order cues, we used identical cues in both cases (a '1' or '2' printed in red or blue) and instructed participants to use cue-colour in half the blocks, while using cue number (order) in the other half of the blocks. In Experiment 2, we again relied on colour cues for several reasons. First, the use of colour cues ensured methodological consistency with Experiment 1, enabling us to build cohesively upon its findings. Second, our previous research has consistently highlighted the effectiveness of colour cues in evoking space-based mnemonic content selection[11,29–32,34,35]. While we also considered directly using spatial cues, this would have had several drawbacks. First, introducing spatial cues would shift the focus away from incidental spatial engagement, which is the study's underlying foundation, and toward direct spatial referencing. Second, when using spatial cues, the gaze biases that were studied here could be directly driven by cue processing rather than reflecting mnemonic space-based selection. Colour cues have the advantage of not causing any spatial biases in gaze directly due to the cue's bottom-up stimulus features. Blocks were randomly interleaved, and the instruction ('use COLOUR' or 'use ORDER') remained printed on the screen throughout the block. Participants completed a total of 480 trials equally spread across 5 colour and 5 order blocks. In total, the experiment lasted ~70 min per participant.

**Data acquisition and preprocessing**. Binocular eye movements were recorded with an EyeLink 1000 Plus (SR Research, Ltd.) at a sampling rate of 1000 Hz. The eye-tracking data were converted into ASCII format and analysed with Matlab and Fieldtrip. Eye blinks were detected and interpolated with a custom spline interpolation procedure. Data were segmented into epochs from −500 to 1500 ms relative to the onset of the retrocue as well as relative to the custom calibration points that we used to normalize the data. To do so, median gaze position (in the period

400–1000 ms post calibration-point onset) was calculated for each of the 7 calibration positions, which were subsequently used to normalize gaze-position data in the task data. We normalized our gaze data such that central fixation was defined as 0, while the centre of the stimulus positions on the left was defined as −100 and the right as +100, corresponding to ±8 degrees visual angle. To correct for any residual central fixation offsets, the resulting epochs were additionally baseline corrected on the 500 ms preceding the retro-cue. It has previously been established that the gaze marker of interest[11, 32] is driven by small microsaccades toward the memorized item location that typically fall well below 50 normalized units (~4 visual degrees). To enhance data sensitivity, we employed two specific exclusion criteria. First, any trial in which the gaze position surpassed 50 normalized units was omitted, identical to the procedure used in ref. [29]. Second, we used participants' response-onset times to estimate their attentiveness to the task and removed trials that had a response onset slower than the mean response onset +4 standard deviations (following an iterative procedure until no more RT-outliers would be left). For Experiment 1, an average of 2.90% of trials (SD = 5.09%) were excluded. Breaking down by specific criteria, due to excessive eye movements, 1.20% (SD = 4.77%) of trials from the sequential condition and 1.39% (SD = 5.20%) from the simultaneous condition had to be removed. Due to inattentiveness, 1.38% (SD = 1.11%) of trials from the sequential condition and 1.92% (SD = 0.85%) from the simultaneous condition were removed. For Experiment 2, an average of 2.81% of trials (SD = 2.69%) were excluded. Specifically, due to excessive eye movements, 1.20% (SD = 2.26%) of trials from the colour condition and 1.45% (SD = 2.93%) from the order condition were removed. As for inattentiveness, 1.44% (SD = 0.90%) of trials from the colour condition and 1.60% (SD = 0.95%) from the order condition were excluded. Taken together, these results confirm that our task did not induce substantial eye movements following the central retrocue.

### Data analysis

*Behaviour.* For the behavioural data analyses we considered two measures: precision and response onset time. Precision was defined as the absolute difference in degrees between the orientation that was reported and the actual orientation of the cued memory item. The response onset time refers to the amount of time it takes to initiate a response after the fixation cross turned grey, which indicated that the participant could provide their response by moving the computer mouse in the direction corresponding with the orientation of the cued memory item.

*Gaze position.* We evaluate the incidental use of space by comparing the horizontal gaze position following a cue for an item that happened to be encoded on either the left or the right side of the fixation cross (note how we never asked participants about the location of cued memory items). First, the gaze position was split based on condition (E1: sequential, simultaneous; E2: colour, order) and the incidental location (left, right) of the cued memory item. Second, to increase sensitivity and interpretability, we followed previous studies[11,29–31] in calculating a standardised 'towardness' metric by contrasting the trial-averaged gaze position time courses (with positive values denoting rightward gaze) in trials in which the cue directed attention to the right minus the left memory item, divided by two.

*Gaze shifts (saccade rate).* In addition to *the continuous* gaze position, we analysed saccadic eye movements. For this, we followed the same steps as in our prior work[32], as described below.

Saccades were detected using a custom procedure of thresholding gaze velocity profiles that was also used and described in ref. [32]. We only considered saccades with a magnitude larger than 1 normalized units (which corresponds to ~0.08 visual degrees). Furthermore, since memory items were exclusively (and deliberately) separated along the horizontal axis, we focused exclusively on gaze shifts along the horizontal axis. To evaluate directionality, saccades were classified as moving towards the target, or moving away from the target. The resulting 'towards' and 'away' saccade time courses (expressed in Hz) were smoothed (using the built-in function "smoothdata" in MATLAB) using a moving average with a 100 ms sliding window. For ease of comparison, we also calculated the saccade effect, as the contrast between 'towards' and 'away' saccades, with positive values indicating a bias toward the target, and negative values a bias away from the target.

*Gaze-shift size (saccade magnitude).* We additionally calculated a time-magnitude representation of gaze shifts (as in ref. [32]), and again contrasted the number of 'toward' and 'away' saccades at each timepoint. To do so, we repeated our gaze-shift analysis, and sorted our data according to gaze-shift sizes ranging from 2 to 110 normalized units, with a bin-size of 4 units. Unlike the previous analyses, in this analysis, we also included trials in which gaze position exceeded 50 normalized units as our aim was to comprehensively visualise the type of saccades that contributed to the overall gaze-position and gaze-shift biases. Only trials that were previously removed due to slow responses were left out.

**Statistical analyses**. In order to assess whether there is a trade-off between the incidental use of space and time for working memory, we compared the behavioural and gaze position data for the two conditions within each experiment. For the statistical evaluation of the two behavioural measures (reproduction errors and response onset times after the memory probe), we used paired samples *t*-tests (two-sided with an alpha level of 0.05), and report Cohen's *d* as a measure of effect size. The data distribution was assumed to be normal but this was not formally tested. The trial-averaged gaze-position and gaze-shift time courses were statistically evaluated using a cluster-based permutation approach[36] that effectively circumvents the multiple-comparisons problem while allowing to evaluate condition differences between two time-courses (without requiring a-priori assumptions regarding specific time windows of interest). We used default configuration settings of the Fieldtrip toolbox[37] to identify significant clusters (10,000 permutations; alpha level of 0.025 per side). We used the BayesFactor package[38] in R for our Bayesian analyses, with default priors ($r = 0.707$). Our analyses relied on Bayes Factors (BF$_{10}$) to quantify the evidence supporting the alternative hypothesis over the null hypothesis, enabling us to assess the evidence for both the presence and absence of effects[39]. We calculated Bayes Factors for towardness and saccade rates using the 'ttestBF' function for Bayesian one-sample *t*-tests, which involved contrasting the averaged values between the two conditions over the entire delay period of each experiment to test the null hypothesis (mu = 0) against an alternative hypothesis suggesting a non-zero effect size ($r = 0.707$). Additionally, we conducted sensitivity analyses to assess the robustness of our results to variations in the prior, which involved recalculating Bayes Factors with different prior scale factors ($r = 0.5, 1,$ and $2$).

**Reporting summary**. Further information on research design is available in the Nature Portfolio Reporting Summary linked to this article.

## Results

**Incidental use of space for working memory is independent of the concurrent availability of time**. The purpose of Experiment 1 (Fig. 2a) was to investigate whether observers rely less on space when memory items can also be individuated through time. To index the incidental use of space for working memory, we tracked horizontal gaze position following internal item selection (as in refs. [11,31]) and compared this when items that had been encoded simultaneously or sequentially to the left and right of fixation. By manipulating the temporal separation between the items we were able to compare conditions in which participants could (sequential) or could not (simultaneous) additionally rely on time for item individuation and subsequent attentional selection and prioritization in visual working memory.

Before turning to our key results regarding the incidental use of space, we checked that task-performance was largely comparable between our two conditions. Figure 2b shows that response onset times were similar for sequential ($M = 224.67$ ms, SEM $= 30.63$ ms) and simultaneous conditions ($M = 238.80$ ms, SEM $= 30.90$ ms), $t(24) = 1.68$, $p = 0.107$, $d = 0.34$, 95% CI $= [-3.279, 31.536]$. Participants were slightly less precise in reporting the orientation (larger reproduction errors) following sequential ($M = 14.28$, SEM $= 1.01$) versus simultaneous encoding ($M = 12.82$, SEM $= 0.88$), $t(24) = -5.80$, $p < 0.001$, $d = 1.16$, 95% CI $= [-1.983, -0.942]$, though they performed clearly

above chance level (corresponding to an average reproduction error of 45 degrees) in both cases.

We now turn to our main results regarding the incidental use of space for working memory. Figure 2c shows the average gaze position ($\pm1$ s.e.m) for sequential (left panel) and simultaneous (right panel) conditions when the colour change of the central fixation cross-cued participants to attend to the item that happened to be either on the left or the right during encoding. Following the onset of the central colour retro-cue (that was itself non-spatial), gaze clearly became biased towards the location where the target was encoded: gaze position became biased towards the left if the probed item previously occupied the space left of the fixation cross, and became biased towards the right for items that were encoded on the right side of the fixation cross. Since participants were never asked about the location, this signified the incidental use of space and shows we can sensitively pick this up with our gaze marker, consistent with previous findings[11,29,31,32].

In order to facilitate the comparison between our sequential and simultaneous conditions, we collapsed the time courses for each condition into a single 'towardness' metric (Fig. 2d). Cluster-based permutation analyses on these towardness time courses confirmed a robust effect, that was clearly present in, and highly similar between, both sequential (cluster $p < 0.001$) and simultaneous (cluster $p < 0.001$) conditions, in each case starting ~300 ms after probe onset, and peaking after ~600 ms.

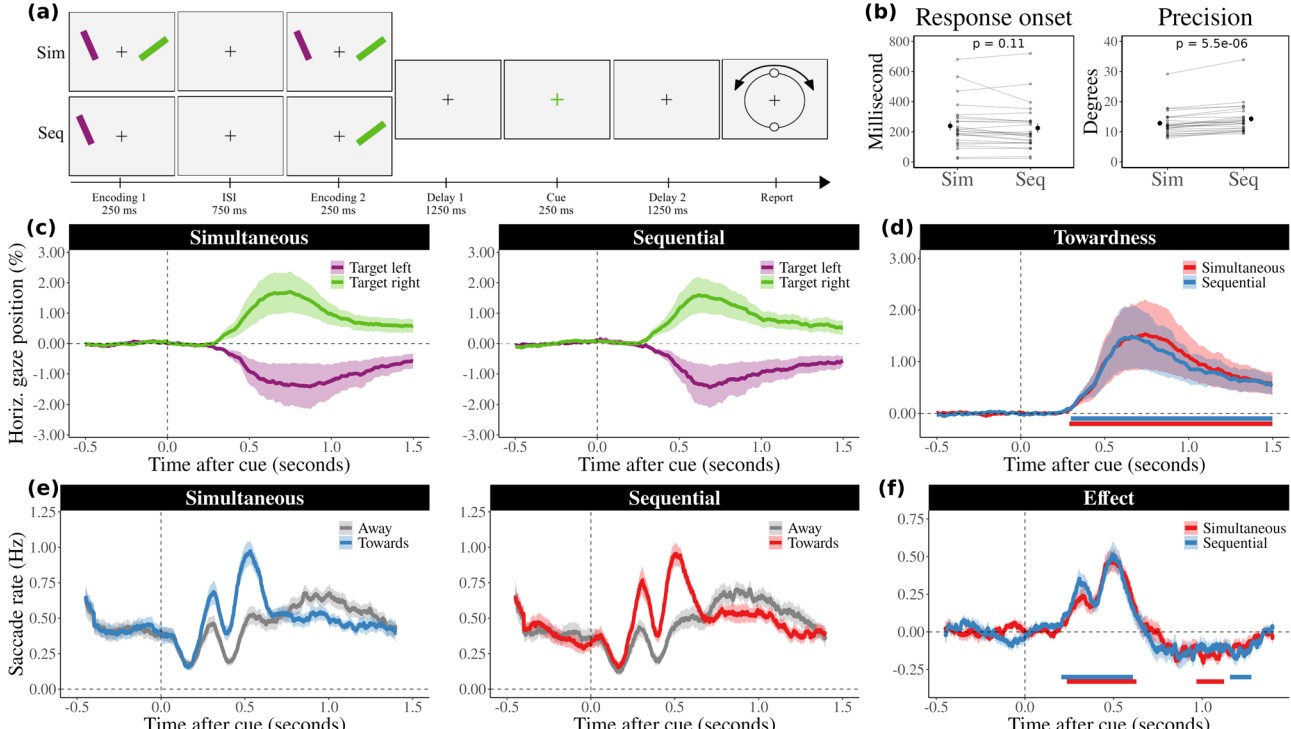

**Fig. 2 Incidental use of space is independent of the concurrent availability of time for item selection and individuation. a** Task schematic of Experiment 1. Participants ($n = 25$) memorized the colour and orientation of two bars and reproduced the orientation of the item that matched the subsequent colour cue. The temporal separation between the two items was varied by presenting items either simultaneously or sequentially. **b** Average accuracy and response onset times for trials with simultaneous and sequential stimulus presentations. The grey lines indicate the performance of individual participants. Group averages are marked to the side ($\pm1$ s.e.m.). **c** Average horizontal gaze position following the retrocue is biased toward the encoding location of the cued memory item. Shaded areas represent 1 s.e.m. The gaze bias is displayed in a normalized scale with a range from $\pm100$ which corresponds to approximately $\pm8$ visual degrees. **d** The same gaze bias but now expressed in a standardized 'towardness' metric that is calculated as the difference in average gaze position in trials in which the cue directed attention to the right minus the left memory item, divided by two. Cluster-based permutation tests revealed that the gaze bias is significant in both simultaneous and sequential conditions, as indicated by the horizontal lines. **e** Time courses of the number of saccades per second. Saccades were classified as moving either towards or away from the encoded location of the cued memory item. **f** The contrast of the number of towards and away saccades confirms an initial bias towards the location of the probed memory item (positive values) which is again similar in both simultaneous and sequential conditions, and is followed by saccades that bring the gaze position back towards the central fixation cross (negative values).

To complement the analysis of gaze position, we additionally evaluated gaze shifts (saccades) as a function of saccade direction (as in ref. [32]). Figure 2e shows that directional biases in saccades follow a similar time course: early saccades were biased towards the encoded location of the target item, followed by an increase in the opposite direction (i.e., back towards the fixation cross). A cluster-based permutation analysis of the time courses confirmed a significant effect of saccade direction that was again similarly present in both sequential (positive cluster $p < 0.001$; negative cluster $p = 0.015$) and simultaneous (positive cluster $p < 0.001$; negative cluster $p = 0.008$) conditions (Fig. 2f).

Direct comparisons between the two conditions showed no statistically significant clusters in the time courses of both towardness and saccade rates. Evaluating the contrast between conditions for average values across the full delay period following the retrocue showed the data were more probable under the null hypothesis ($t(24) = -0.77$, $p = 0.449$, $d = 0.15$, 95% CI = [−0.206, 0.094], $BF_{10} = 3.62$ for towardness, and $t(24) = 0.15$, $p = 0.883$, $d = 0.03$, 95% CI = [−0.018, 0.021], $BF_{10} = 4.70$ for saccade rates). These Bayes Factors indicate that the data are about 3.62 and 4.70 times more likely under the null hypothesis than under the alternative hypothesis, respectively. Therefore, our results suggest that there is little credible evidence for a difference between the two conditions in terms of both towardness and saccade rates. Furthermore, our sensitivity analysis, which recalculated Bayes Factors with prior scales of 0.5, 1, and 2, showed consistent favour towards the null hypothesis. The $BF_{10}$ values for towardness were 2.76, 4.89, and 9.38, and for saccade rates, they were 3.51, 6.43, and 12.51. This

consistency across all prior scales underscores the resilience of our findings against varying prior assumptions, reinforcing the credibility of our conclusions.

Taken together, these data show that space—as quantified by our spatially indexed gaze bias measure—is similarly used for item individuation and selection in working memory, regardless whether items could also be individuated by time (sequential conditions) or not (simultaneous condition).

**Space remains important for working memory even when participants are forced to use time.** In Experiment 1, we varied the availability of time, for individuating and selecting two items in visual working memory. Our gaze marker showed how the incidental use of space is independent of the concurrent availability of time for item selection and individuation. However, in Experiment 1, participants could but did not need to, rely on time for item individuation and selection.

The purpose of Experiment 2 (Fig. 3a) was, therefore, to investigate the incidental use of space for item individuation and selection when participants were vs. were not *forced* to use time to complete the task. To achieve this, we always presented the two items sequentially but this time cued the relevant memory item either through its colour (as in Experiment 1) or its temporal order (first or second) at encoding. We did so while using the exact same cues ('1' or '2' printed in the colour of either item) and only varying the instructions whether to use cue-colour or cue-number (order). Our analyses for Experiment 2 followed the same logic as in Experiment 1.

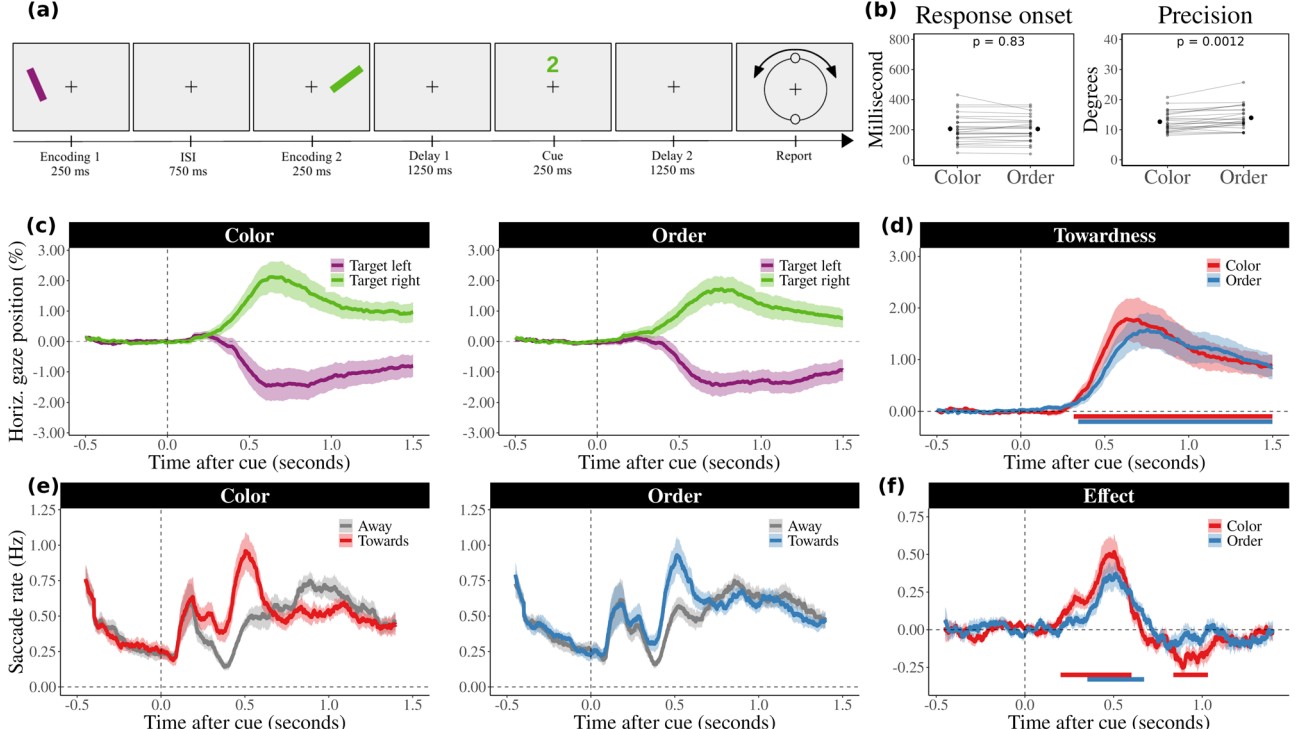

**Fig. 3 Preserved incidental use of space for visual working memory even when forced to individuate items across time. a** Task schematic of Experiment 2. This experiment comprised a working memory task that was largely the same as in Experiment 1, only now the two items were always presented sequentially, and participants (n = 25) were cued either through the cue-colour (as in Experiment 1) or through cue-number (temporal order; first or second). This example shows a trial where in the colour block participants would have to select the green (second) item, whereas in the order block participants would have to select the first (purple) item. **b** Average accuracy and response onset times for the two conditions. The grey lines indicate the performance of individual participants. Group averages are marked in red (±1 s.e.m.). **c** The average horizontal gaze position following item selection was again biased towards the encoding location of the probed memory item. Shaded areas represent 1 s.e.m. **d** Cluster-based permutation tests revealed that the gaze bias is significant in both conditions, as indicated by the horizontal lines. **e** The time courses of the saccade rates show a pattern that is similar as in Experiment 1. **f** The contrast of the number of towards and away saccades also follows a similar pattern as in Experiment 1.

We again first checked that performance was largely comparable between the two conditions. Figure 2b shows that response onset times were similar following colour ($M = 205.77$ ms, SEM = 18.76) and order cues ($M = 204.63$ ms, SEM = 16.86), $t(24) = 0.22$, $p = 0.83$, $d = 0.04$, 95% CI = [−9.581, 11.857], but that participant were slightly less precise in reporting the orientation of items following order cues ($M = 13.95$ degrees, SEM = 0.79) than colour cues ($M = 12.65$ degrees, SEM = 0.67), $t(24) = −3.66$, $p = 0.0012$, $d = 0.73$, 95% CI = [−2.027, −0.566] (Fig. 3b). However, again, performance was well above chance performance in both cases.

Next, we turn to our main results regarding the use of space, as again indexed by our gaze marker. Figure 3c shows the average horizontal gaze position (±1 s.e.m.) for trials in which the relevant memory item was cued by the colour (left panel) or serial order (right panel). After the onset of the retro-cue, we observed a similar pattern in gaze position as in Experiment 1: gaze position became biased towards the left if the probed item previously occupied the space left of the fixation cross, and became biased towards the right for items that were encoded on the right side of the fixation cross. Cluster-based permutation analyses of the 'towardness' time courses again showed a robust effect in both conditions (cluster $p < 0.001$ in the cue-colour condition, and cluster $p < 0.001$ in the cue-order condition), starting around 300 ms after probe onset, and peaking after 600 ms (Fig. 3d).

We again complemented the gaze-position analyses, with an evaluation of the directionality of the gaze shifts (saccades). Figure 3e shows that saccades follow a similar time course: early saccades were biased towards the encoding location of the target item, followed by an increase in the opposite direction (back towards the fixation cross). A cluster-based permutation analysis of the time courses showed a significant effect of saccade direction in both conditions (Fig. 3f, positive cluster $p < 0.001$, negative cluster $p = 0.004$ in the cue-colour condition; cluster $p < 0.001$ in the cue-order condition). This effect appeared to be slightly larger (albeit not significantly) in the color conditions than it did in the order condition. Critically, however, it was clear and robust in both cases showing that the incidental use of space remains profound even when participants are forced to individuate working-memory items across time (order-cue condition).

Mirroring Experiment 1, direct comparisons between conditions showed no significant clusters in the time courses of towardness and saccade rates. Evaluating the contrast between conditions for average values across the full delay period after the retrocue showed that the data are more probable under the null hypothesis ($t(24) = 0.44$, $p = 0.666$, $d = 0.09$, 95% CI = [−0.199, 0.306], $BF_{10} = 4.35$ for towardness, and $t(24) = 0.09$, $p = 0.932$, $d = 0.02$, 95% CI = [−0.022, 0.024], $BF_{10} = 4.73$ for saccade rates). These Bayes Factors indicate that the data are about 4.35 and 4.73 times more likely under the null hypothesis than under the alternative hypothesis, respectively. Therefore, our results suggest that there is little credible evidence for a difference between the two conditions in terms of both towardness and saccade rates. Additionally, our sensitivity analysis recalculated Bayes Factors using prior scales of 0.5, 1, and 2 and consistently indicated a preference for the null hypothesis. The $BF_{10}$ values for towardness were 3.27, 5.93 and 11.48, and for saccade rates, they were 3.53, 6.48, and 12.61. This consistency across all prior scales underscores the resilience of our findings against varying prior assumptions, reinforcing the credibility of our conclusions.

**The reported spatial gaze biases are driven by directional biases in microsaccades.** In both experiments, we found clear spatial biases in gaze position that were paralleled by biases in the direction of gaze shifts. To further characterize the nature of these biases, we additionally evaluated the differences in the toward and away saccade time courses as a function of saccade size (as in ref. [32]). For this analysis, we included trials with gaze-shifts of all sizes (including those >50% normalized units).

The results for both our experiments are visualized in Fig. 4. These data reveal how, the reported gaze biases are driven predominantly by gaze shifts lower than 1 degree visual angle, with similar patterns across our conditions and across our two experiments. In contrast, we found no evidence for directional biases in gaze shifts that revisited the originally encoded locations of the cued memory items (100% in our analysis). This is in line with a directional bias in microsaccades (as in refs. [40,41], here reported when attention is directed internally (as in refs. [11,29–32]).

## Discussion

We investigated whether there is a trade-off between the incidental use of space and time for visual working memory. Specifically, we asked whether human observers would rely less on space when objects could (Experiment 1) or had to (Experiment 2) additionally be individuated through time (here operationalized as sequential temporal order). Our results suggest that space remains a profound organizing principle that serves the individuation and selection of visual objects from working memory even when time is also available, or even has to be used —with no evidence for an obligatory trade-off between the incidental use of space and time.

We tracked the incidental use of space through spatial biases in gaze when participants directed their attention to objects in visual working memory following a non-spatial retrocue. This revealed that attentional selection from working memory relies on memorized spatial object locations, even when the encoded location of the relevant memory object was never asked for, and nothing was ever expected at that location (the response-dial was always centered around the central fixation cross). While this spatial gaze bias was demonstrated in several prior studies[11,29,31,32]; the unique insight here is that incidental use of space remained profound even when items never competed for space at encoding (sequential encoding), or when memory objects were cued through encoding order (enforcing object individuation through time). These results thus show that, even under such circumstances, space continues to serve as a scaffold for binding mnemonic content in visual working memory to serve later selection and prioritization. This is generally consistent with sensory-recruitment models of visual working memory, that posit that working memory recruits overlapping sensory areas that initially encode the information[13–16,42], with non-spatial object features, such as color and temporal position, theorized to be bound together via their shared position[3–6,18,43–45].

Our data do not argue against the important role that time itself may have for individuation and selection in working memory[20,27,28]—nor the important role for temporal expectations in guiding prioritization[46–49] and protection[50] in working memory. Instead, we investigated whether there is a trade-off between the incidental use of space (spatial separation) and time (temporal separation) and not which is more important—time or space.

Our data are consistent with prior studies that found that the spatial location of sequentially encoded objects continues to affect performance. For example, objects that are encoded sequentially at the same versus different locations are more likely to be judged as having the same identity or features[45,51]. Moreover, recall is better for memory items that were successively presented at unique locations, than for items that were encoded at the same location[3,52,53]. It has even been suggested that spatial crowding may occur similarly in working memory, regardless whether

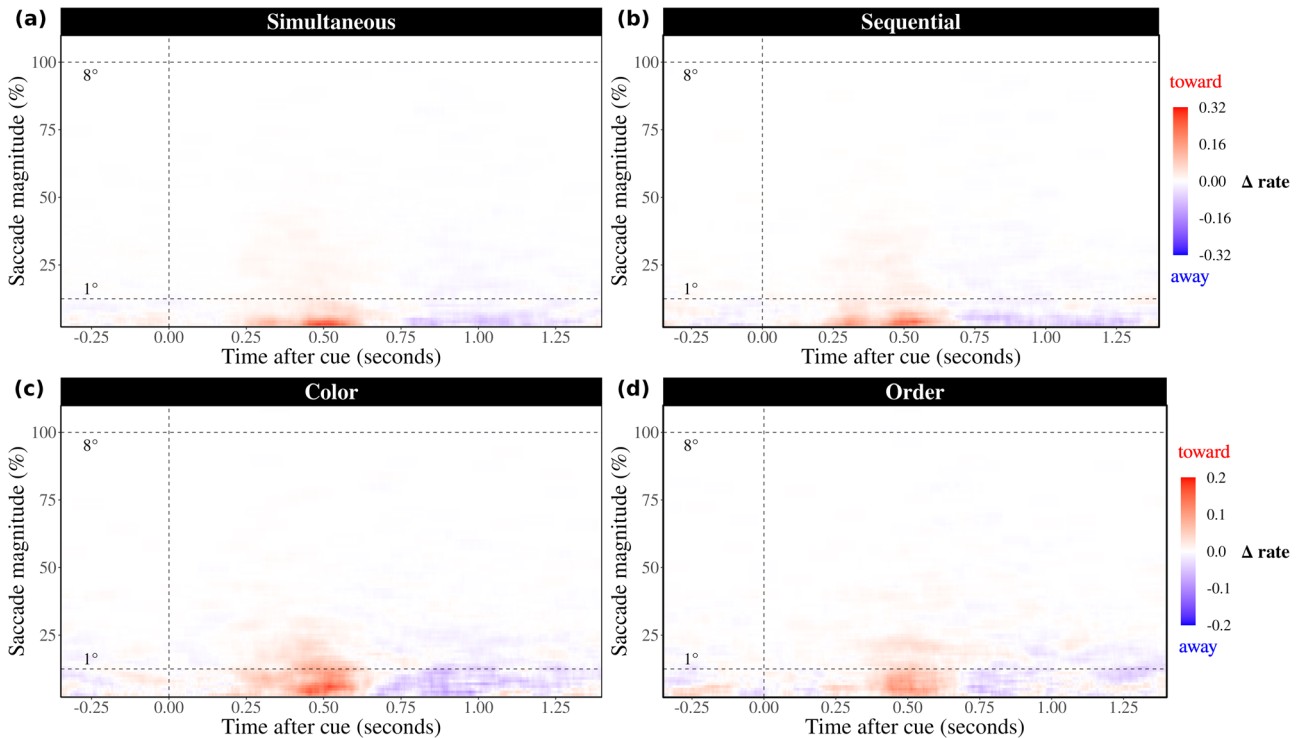

**Fig. 4 Gaze biases towards the location of the cued memory item is driven by microsaccades in all reported conditions.** Panels depict data from Experiments 1 and 2, with each involving $n = 25$ participants: **a** Simultaneous condition in Experiment 1, **b** Sequential condition in Experiment 1, **c** Color condition in Experiment 2, and **d** Order condition in Experiment 2. In each panel, time courses depict the difference in saccade rates (towards minus away) as a function of saccade magnitude. The upper dashed horizontal line indicates the centre location of the cued memory item at encoding (100 normalized units, or 8°), while the lower dashed line marks a distance of 1° relative to the central fixation cross. In all conditions, and in both reported experiments, the gaze bias is driven by an increase in toward saccades of which the vast majority occur below 1°, consistent with a directional bias in microsaccades.

objects are encoded simultaneously, or sequentially[54]. Thus, spatial separation plays an important role in distinguishing between mnemonic content, even when encoded sequentially. Our data show that this is the case even when participants are *required* to retain sequential order.

Recent studies have uncovered the role of microsaccades in various cognitive functions, including their decreased rates with increased memory loads[55,56], prolonged inhibition following the presentation of task-relevant oddball stimuli[57], and variations in rates and magnitudes with mental arithmetic task difficulty[58,59]. Complementing this work on microsaccade rates, our study specifically focused on spatial biases in the direction of microsaccades[40,41,60,61], building on other recent studies demonstrating that such spatial biases are also observed when shifting attention among the contents of visual working memory[11,29–32,35]. As discussed previously[11], this pattern of results differs from related findings on 'looking-at-nothing'[62–68]. In this literature, eye movements revisit the (now empty) location of a relevant object which may facilitate the retrieval of information that was originally encoded at this location. In contrast, in our study participants did not look back at the location of the selected memory object, but showed an increased propensity for their ongoing microsaccades to be biased in this direction. These microsaccade biases may signal the allocation of attention, consistent with studies of microsaccades as a marker of peripheral covert attention[40,41], and demonstrations of peripheral (extrafoveal) consequences of foveal microsaccades[69].

Whereas a number of factors may account for differences in the pattern of eye movements observed across distinct memory paradigms, we note how both types of oculomotor manifestations can unveil spatial coding in memory. Indeed, complementary to our focus on microsaccades to study spatial coding in visual working memory, related studies have used looking-at-nothing to study the role of spatial coding in memory and retrieval[20,70] and for studying the hierarchy of spatial and temporal memory frames, following explicit instruction to retain spatial vs. temporal relations in working memory[20].

In the current study, we demonstrate that space remains a profound organizing principle for working memory, even when objects are additionally individuated through time. We have shown this while the spatial relations between memoranda and observer remained fixed, as is typical in laboratory investigations of this type. A recent study[31] used virtual reality to demonstrate that everyday working memory following self-movement may incidentally rely on multiple, complementary spatial frames, such as pertaining to where objects are in the world relative to ourselves and relative to each other[71–74]. Thus, in everyday behaviour, the scaffolding role of space in working memory may be even richer than we studied here, serving as a multifaceted organizing principle for the contents of visual working memory. Our data make clear that this use of space remains important, even when time can, or even has to, additionally be used for object individuation in working memory.

**Limitations**. By showing preserved incidental use of space for mnemonic selection when temporal order information was additionally available (Experiment 1) or necessary (Experiment 2) for selection, our data provide proof-of-principle that the use of time for mnemonic selection does not obligatorily come at a cost to the reliance on space. In other words, we show that a trade-off between the use of space and time is not inevitable. At the same time, we do not wish to claim that such a potential trade-off will never occur—we merely show that it does not *necessarily* occur.

In our experiments, memory objects were consistently presented in distinct hemifields, with clear spatial separation, making space a viable and powerful means to individuate the objects, even when subsequently cued through temporal order. It is conceivable that the use of space may be less prominent when the two objects were to be presented within the same hemifield or closer in proximity, or even temporally separated at an identical spatial location. One potential factor influencing the utilization of space or time for object individuation could be the degree of separability between the objects in either dimension, given that decreased separability may lead to heightened inter-item interference. As such, as spatial separation decreases, the cognitive system may be more inclined to use alternate individuating dimensions, such as time, to minimize interference[21,75,76]. Exploring the incidental uses of space and time—and potential trade-offs between them—by systematically varying the separability of the objects will thus be an interesting avenue for subsequent research. Moreover, in future work, neuroimaging can be used, in addition to eyetracking, to provide relevant complementary findings about the neural bases of such potential trade-offs (for a recent example, see ref. [77]).

## Data availability

Raw data has been made publicly available via the Open Science Framework and can be accessed at https://doi.org/10.17605/OSF.IO/86ATZ[78].

## Code availability

The analysis code is available via the Open Science Framework at https://doi.org/10.17605/OSF.IO/86ATZ. The code was implemented using Matlab version R2020a with the FieldTrip toolbox version 20220628, as well as RStudio version 2022.7.2.

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

## Acknowledgements

This research was supported by an ERC Starting Grant from the European Research Council (MEMTICIPATION, 850636) to F.v.E. The funder had no role in study design, data collection and analysis, decision to publish or preparation of the manuscript. The authors wish to thank Heleen Slagter for her valuable comments on the manuscript, as well as Tess Röder for stimulating discussions and literature research in the early stages of this project.

## Author contributions

E.d.V.: Conceptualization, data curation, formal analysis, methodology, project administration, software, visualization, writing—original draft, writing—review & editing. F.v.E.: Conceptualization, funding acquisition, methodology, software, supervision, writing—review & editing. G.F.: Investigation, project administration.

## Competing interests

The authors declare no competing interests.
