## [Peer Review File · Communications Psychology]

22nd Jun 23

Dear Mr de Vries,

Thank you for your patience during the peer-review process. Your manuscript titled "No trade-off between the use of space and time for working memory" has now been seen by 3 reviewers, whose comments are appended below. You will see that they find your work of some potential interest. However, they have raised quite substantial concerns that must be addressed. In light of these comments, we cannot accept the manuscript for publication, but would be interested in considering a revised version that fully addresses these serious concerns.

We hope you will find the Reviewers' comments useful as you decide how to proceed. Should additional work allow you to address these criticisms, we would be happy to look at a substantially revised manuscript. If you choose to take up this option, please highlight all changes in the manuscript text file, and provide a detailed point-by-point reply to the reviewers.

Addressing this concerns requires the collection of new data from an experiment that makes the spatial information more subtle (e.g., presenting items in the same hemifield) and/or less consistently available (e.g., having some spatial-only trials, some temporal-only trials, and some both-available trials). For this reason, we encourage you to integrate the thoughtful suggestions of Reviewer 1 & 3 into the design of a new experiment that would have the potential to satisfy their concerns and provide more powerful evidence for your central claim. We recognize that collecting more data will take time, so we are happy to give you five months to pursue this.

Please align your statistics reporting and interpretation with our guidelines, which you can find here: <https://www.nature.com/commpsychol/submit/submission-guidelines#statistical-guidelines>

For any interpreted null-result, we require Bayes Factors or equivalence tests that yield positive evidence for the null. Please also use appropriate language to describe the results. (There is no statistical test that can demonstrate absence of an effect. Statements such as 'There is no difference between x and y.' or 'X does not affect Y.' must be revised to read 'We found [no/little] credible evidence of a difference between x and y.' or 'We found [no/little] credible evidence that X affects Y.')

Reviewer 2 asks for a more solid justification for the sample size. Please note that although we ask for clear justification for the chosen sample size, we are not asking for post hoc power calculations. Instead, please provide a sensitivity analysis. The reviewer also suggests a number of references. These are valuable suggestions and we encourage you to consider these articles; the degree to which you incorporate the citations is at your discretion.

If the revision process takes significantly longer than five months, we will be happy to reconsider your paper at a later date, provided it still presents a significant contribution to the literature at that stage.

Please use the following link to submit your revised manuscript, point-by-point response to the Reviewers' comments with a list of your changes to the manuscript text (which should be in a separate document to any cover letter) and any completed checklist:

[link redacted]

Please do not hesitate to contact me if you have any questions or would like to discuss the required revisions further. Thank you for the opportunity to review your work.

Best regards,

Jesse Rissman

Jesse Rissman, PhD
Editorial Board Member
Communications Psychology
orcid.org/0000-0001-8889-5539

EDITORIAL POLICIES AND FORMATTING

Editorial Policy: [Policy requirements](https://www.nature.com/documents/nr-editorial-policy-checklist.pdf) (Download the link to your computer as a PDF.)

Furthermore, please align your manuscript with our format requirements, which are summarized on the following checklist:

[Communications Psychology formatting checklist](https://www.nature.com/documents/commspsychol-style-formatting-checklist-article-rr.pdf)

and also in our style and formatting guide [Communications Psychology formatting guide](https://www.nature.com/documents/commspsychol-style-formatting-guide-accept.pdf) .

* **CODE AVAILABILITY:** All Communications Psychology manuscripts must include a section titled "Code Availability" at the end of the methods section. In the event of publication, we require that the custom analysis code supporting your conclusions is made available in a publicly accessible repository; please choose a repository that provides a DOI for the code; the link to the repository and the DOI must be included in the Code Availability statement. Publication as Supplementary Information will not suffice. We ask you to prepare and upload code at this stage, to avoid delays later on in the process.

* **DATA AVAILABILITY:**

All Communications Psychology research manuscripts must include a section titled "Data Availability" at the end of the Methods section or main text (if no Methods). More information on this policy, is available at <http://www.nature.com/authors/policies/data/data-availability-statements-data-citations.pdf>.

At a minimum the Data availability statement must explain how the data can be obtained and whether there are any restrictions on data sharing. Communications Psychology strongly endorses open sharing of data. If you do make your data openly available, please include in the statement:

We recommend submitting the data to discipline-specific, community-recognized repositories, where possible and a list of recommended repositories is provided at <http://www.nature.com/sdata/policies/repositories>.

If a community resource is unavailable, data can be submitted to generalist repositories such as [figshare](https://figshare.com/) or [Dryad Digital Repository](http://datadryad.org/). Please provide a unique identifier for the data (for example a DOI or a permanent URL) in the data availability statement, if possible. If the repository does not provide identifiers, we encourage authors to supply the search terms that will return the data. For data that have been obtained from publicly available sources, please provide a URL and the specific data product name in the data availability statement. Data with a DOI should be further cited in the methods reference section.

Please refer to our data policies at http://www.nature.com/sdata/policies

<http://www.nature.com/authors/policies/availability.html>

Reviewer expertise:

Reviewer #1: working memory

Reviewer #2: eye tracking

Reviewer #3: visual working memory

Reviewer #1 (Remarks to the Author):

This manuscript describes two behavioral studies on spatial reliance in visual working memory. Previous studies have shown that even when location is task-irrelevant, people use space to individuate items in WM. The current study asked if this reliance is preserved or reduced if the WM items can also be individuated by temporal cues. In the baseline Simultaneous task, two WM items (colored tilted bars) were presented simultaneously, then during the WM delay, a central cue indicated the color of the to-be-probed item; after an additional delay, subjects reported the orientation of that remembered item. The key measure is eye position gaze bias during the second delay: the authors replicate prior studies showing that gaze position is biased toward the location of the to-be-probed item, in the absence of visual input. In Experiment 1, the authors compare this “towardness” bias to a Sequential condition, where the 2 WM items are presented sequentially, at different points in time; they report similar gaze bias for sequential and simultaneous conditions. In Experiment 2, all items are presented sequentially, and the to-be-probed item is cued either by its color or its temporal position; again, the authors report similar gaze bias for these conditions. The conclusion is that location reliance is automatic and there is no tradeoff when temporal cues are available.

I thought this was an interesting paper asking a neat and novel question. I liked the studies, though I’m left feeling not quite convinced about the conclusion. To be clear, I think the results definitively show a strong reliance on location information in these scenarios where location is increasingly not needed, and I think that could make a nice contribution to the literature. I’m just less convinced that the “no trade-off” claim is being compellingly tested.

Major concerns:

1. The study aims to test a tradeoff between the use of space and time. But the way the cues are instantiated in the current study doesn’t feel like they are equally strong, from a methodological perspective. For example, space here is always a coarse hemifield distinction. The items are always presented on opposite sides of the hemifield. This is a very obvious spatial cue, with the items encoded by different hemispheres of visual cortex, and that could contribute to the robust/automatic nature of this effect. On the other hand, the temporal cues are of weaker and more relative magnitude (across the entire 1-hr experiment, the temporal differences between the two items on a given trial only differ by 1 second. The spatial tags are 2 consistent, absolute locations across the entire experiment, while the temporal tags are changing across the whole experiment, which unfolds in time. I think this needs discussing, and serious consideration of whether the paradigm allows for a fair testing of a potential “tradeoff”.

2. Another imbalance is that in Experiment 1, spatial cues are always present and available to individuate, while temporal cues are only available on half of the trials. It feels like a better test would be to have some spatial-only trials (simultaneous), some temporal-only trials (sequential at

same location), and some both-available trials (sequential at different locations). The key comparison would still be gaze bias in spatial-only vs both-available, but in this context, space can't always be used, so it seems like a fairer test of the tradeoff.

3. The hypothetical experiment described above would also allow for another stronger test of the automaticity of spatial indexing: Would gaze still be biased toward the item's location if location could NOT be used to differentiate? Imagine the condition where the two items are presented sequentially, both on the left side. Then one of the items is cued by either color or temporal position. Would gaze still be biased toward the left side? If so, that supports an automatic, sensory recruitment explanation, regardless of whether space can be used for individuation.

4. I appreciate the motivation for Experiment 2, having a condition that explicitly uses temporal order to cue the to-be-reported item. But the experiment doesn't seem to be directly testing a space-time tradeoff, more of a color-time tradeoff. This is not to say I don't find the existence of the spatial bias in the order condition compelling. But using it as evidence for "no tradeoff between the use of space and time" (title) feels like too much of an over-reach.

5. The statistical tests were focused on when each condition individually was significant from zero. But unless I missed it, I didn't see any direct comparisons between conditions. While it's clear there is a strong bias in all conditions, were there any differences in the towardness metric, e.g. in terms of amplitude or duration? In Experiment 1, it looks like there's a consistently stronger and more prolonged bias in the simultaneous condition, and in Experiment 2, the bias in the color condition appears to peak earlier and higher.

Reviewer #2 (Remarks to the Author):

This is a nice paper in which two experiments aimed at revealing the possible relationship between space and time in WM are reported. Both experiments relied on a simple task in which the position or the colour of the stimulus had to be memorised while eye movements were recorded. In summary, space appears to be an important dimension for WM even when other information could be used to complete a task.

I have found this work to be well-organised and written and I have little to say.

My main comment is related to sample size: I think a subjective criterion (e.g., $N = 25$ because other studies have done something similar) could open several questions and a power analysis is always preferable. So, I am wondering if more details can be added to provide a more solid justification of the priori-established sample size used in both experiments.

My second comment is about the coverage of the literature. In recent years microsaccades have been 'rediscovered' and in particular their relationship with higher cognitive mechanisms. In this regard, I am missing some studies reporting a link between these tiny eye movements and working memory, operationalised at different levels and with different tasks, such as mental arithmetic (Siegenthaler et al., 2014) or counting (Valsecchi et al., 2007), or with Sternberg's task (Dalmaso et al., 2017). I think it would be fair for these studies to be added and briefly commented on in the general discussion.

References

- Dalmaso, M., Castelli, L., Scatturin, P., & Galfano, G. (2017). Working memory load modulates microsaccadic rate. *Journal of Vision*, 17(3), 6. <https://doi.org/10.1167/17.3.6>
- Siegenthaler, E., Costela, F. M., Mccamy, M. B., Di Stasi, L. L., Otero-Millan, J., Sonderegger, A., Groner, R., Macknik, S., & Martinez-Conde, S. (2014). Task difficulty in mental arithmetic affects microsaccadic rates and magnitudes. *European Journal of Neuroscience*, 39(2), 287–294. <https://doi.org/10.1111/ejn.12395>
- Valsecchi, M., Betta, E., & Turatto, M. (2007). Visual oddballs induce prolonged microsaccadic inhibition. *Experimental Brain Research*, 177(2), 196–208. <https://doi.org/10.1007/s00221-006-0665-6>

Reviewer #3 (Remarks to the Author):

COMMSPSYCHOL-23-9116-T Review

SUMMARY

This manuscript examines the role of space and time in visual working memory – specifically whether observers will alleviate their use of space when separation in time allows object individuation. In the first experiment, two colored bars with varying orientations are presented simultaneously (two at a time; sometimes early, late or both early and late timepoints) or sequentially (one at a time; left then right, or vice versa). Participants were cued to an item with a color and required to report the orientation of that item. In the second experiment, the items were always presented sequentially, but the retro-cue was either a color or the temporal order (first or second). The authors track microsaccades towards a cued item as an operationalization for use of space (spatial gaze bias) and find that spatial gaze biases remain unchanged despite the inclusion of temporal individuation, suggesting that there is no trade-off of time and space, and that space is a strong organizational property for working memories.

I think the issue at hand is very current and critical to the study of visual working memory – the empirical results are likely of interest to the wider field. However, I am unconvinced that the manipulations employed in the presented experiments ‘forced’ participants to retrieve via a temporal index in a way that would be at the cost of the spatial index. That is, I’m not sure how far the empirical finding will generalize. I hope the authors find the following comments constructive in improving their manuscript.

COMMENTS

1. I commend the authors for making their data and code openly accessible on the Open Science Framework – I accessed these files to inform my review. I also confirmed using statcheck.io that the t-statistics and p-values reported in the manuscript are consistent.

a) The repository could use some README files to explain the variables in the raw data .txt files, as well as a brief explanation of the different functions attached. This will increase the reproducibility of the studies presented in the manuscript.

b) Could the authors also upload the 'beh_subject_avg.csv' file or the function for the iterative exclusion procedure of slow response times?

I tried to reproduce the behavioral analyses (paired t-tests on response onset times and recall accuracy) using the 'behavior_all_subjects.csv' in the 'processed_data' folder, before realizing that the processed data file likely contains all trials before exclusion due to slow response onset times.

Doing this re-analysis revealed that most reported effects were robust to data cleaning procedures – the one result differing to what was reported in the manuscript was the response onset times for simultaneous versus sequential presentations before trial exclusions (significantly slower for simultaneous than sequential). This suggests that the simultaneous condition elicited more extremely delayed responses – perhaps without the temporal index available, retrieval of the feature value via the spatial index is slower, but with it available, retrieval is speeded and lowers response onset times. The proportion of excluded trials should be reported by condition within each experiment, rather than by experiment (page 9, line 420). I have attached my reanalysis as a .RMD file and .html file for reference.

2. The authors suggest that because there was no difference in spatial gaze bias when retro-cued with a temporal index compared to with the color of the item (i.e. spatial organization is still used with temporal indexing) that “the use of space remains important, even when time can, or even has to, additionally be used for object individuation in working memory” (page 7, line 331). I don't think this has been definitively shown yet – in my view, the experimental conditions do not impose a significant enough load where spatial organization may be lost. It is interesting that participants retain spatial organization of the items when retro-cued with a temporal index, but this may be because that spatial organization helps boost retrieval of the featural value and there is no impetus to drop the spatial organization. One might expect that when the discriminability of spatial index is difficult (objects are presented close together) that the use of spatial information may be less relied upon (and the temporal index may be more relied upon, though to my knowledge we don't have a measure for this reliance).

A potential follow-up experiment that might be more convincing would be to include trials where the spatial organization hinders the retrieval of the featural value – say, if items are presented on the same side (similar spatial locations) forcing subjects to better individuate through temporal order. Participants' spatial gaze bias may be suppressed as a result of the spatial index being confusing. This new condition would likely need to include some spatial placeholders to keep the visual center of mass around the fixation cross, as was done in the experiments presented so far. Comparing the degree of spatial bias in those conditions compared to when the items are presented at clear separate spatial locations (left and right) as was done here would be a more definitive test. Observing a spatial gaze bias that is no different would then be a more convincing demonstration that there is no trade-off.

A useful framework for the present set of studies is one proposed by the Interference Model (Oberauer and Lin, 2017). They propose that similarity in the context (the feature dimension cued

for retrieval) will produce more interference, resulting in more errors in reporting the correct feature value. It is likely that the observer may trade-off between the spatial and temporal conditions when discriminability in one of those dimensions is low and the other is high. Here, the present studies only test cases when spatial discriminability is high, perhaps masking any effect of trade-off. Note that Schneegans, McMaster and Bays (2022) in similar experiments find both location and ordinal (temporal) position mediate binding and have similar contributions to error correlations.

Oberauer, K., & Lin, H. Y. (2017). An interference model of visual working memory. *Psychological review*, 124(1), 21.

Schneegans, S., McMaster, J., & Bays, P. M. (2022). Role of time in binding features in visual working memory. *Psychological Review*.

Some minor typos:

(page 7, line 292) missing space after the bracket.

(Page 3, line 150) “pick this us” rather than “pick this up”

William Xiang Quan Ngiam

I sign all my reviews, regardless of the recommendation to the editor. By signing this review, I affirm that I have made my best attempt to be polite and respectful while providing criticism and feedback that is hopefully helpful and reasonable. For transparency, my peer review process is detailed at https://williamngiam.github.io/blog/my_peer_review_process.

I completed this review with help from Henry Jones, a graduate student from the same lab.

```

---
title: "COMMSPSYCHOL-23-9116 REVIEW"
output: pdf_document
---
Reproduction attempt of statistical analyses in review of COMMSPSYCHOL-23-9116 ("No trade-off between the use of space and time for working memory")

It is likely that analyses don't match what is reported in the manuscript because the provided data file includes trials that were excluded from the analysis due to slow response onset times.

Written by William X.Q. Ngiam (wngiam@uchicago.edu)

```{r setup, message=FALSE, warning=FALSE}
Load packages
library(tidyverse)
library(dplyr)

Markdown output style option
knitr::opts_chunk$set(comment=NA)

Load data
exp_data <- read_csv('/Users/will.ngiam/Desktop/behavior_all_subjects.csv')
```

## Experiment 1 (Simultaneous versus Sequential)

### Calculate descriptive statistics

```{r Experiment 1 Descriptives, message=FALSE, warning=FALSE}
Create data frame for Experiment 1
exp_1_data <- exp_data %>%
 group_by(experiment, subject, condition) %>%
 filter(experiment == 1) %>%
 summarise(error = mean(reportVsTarget), rt = mean(RT1))

Descriptive stats
aggregate(exp_1_data$error, list(exp_1_data$condition), FUN = mean)
aggregate(exp_1_data$error, list(exp_1_data$condition), FUN = sd)

aggregate(exp_1_data$rt, list(exp_1_data$condition), FUN = mean)
aggregate(exp_1_data$rt, list(exp_1_data$condition), FUN = sd)
```

### Perform paired t-tests

The t-test for the average report error and the average reaction time is slightly different to what is reported in the manuscript.

From the manuscript: "Figure 2b shows that response onset times were similar for sequential (M = 224.67 ms, SEM = 30.63 ms) and simultaneous conditions (M = 238.80 ms, SEM = 30.90 ms),  $t(24) = -1.68$ ,  $p = 0.107$ ,  $d = 0.34$ ."

```

I find the paired t-test for reaction times is different. There is a

significant difference between reaction times in the sequential versus simultaneous conditions (significantly slower in the simultaneous condition)

```
```{r Experiment 1 RT test, warning=FALSE, message=FALSE}
t.test(rt ~ condition, data = exp_1_data, paired = TRUE, alternative =
"two.sided")
```
```

From the manuscript: "Participants were slightly less precise in reporting the orientation (larger reproduction errors) following sequential ($M = 14.28$, $SEM = 1.01$) versus simultaneous encoding ($M = 12.82$, $SEM = 0.88$), $t(24) = 5.80$, $p < 0.01$, $d = 1.16$, though they performed clearly above chance level (corresponding to an average reproduction error of 45 degrees) in both cases."

I also find that the errors were significantly larger in the sequential condition compared to the simultaneous condition, but the t-statistic is a little smaller ($*t* = 4.86$).

```
```{r Experiment 1 recall error test, warning=FALSE, message=FALSE}
t.test(error ~ condition, data = exp_1_data, paired = TRUE, alternative =
"two.sided")
```
```

Experiment 2

Calculate descriptive statistics

The means and standard deviations for the report error and the response onset times are different to what was reported in the manuscript.

From the manuscript, the mean response onset time follow color cues is reported as 205.77ms and following order cues is 204.63ms. I find that it is 246.67ms following color cues and 244.61ms following order cues.

```
```{r Experiment 2 Descriptives, message=FALSE, warning=FALSE}
exp_2_data <- exp_data %>%
 group_by(experiment, subject, condition) %>%
 filter(experiment == 2) %>%
 summarise(error = mean(reportVsTarget), rt = mean(RT1))
```

# Descriptive stats

```
aggregate(exp_2_data$error, list(exp_2_data$condition), FUN = mean)
aggregate(exp_2_data$error, list(exp_2_data$condition), FUN = sd)
```

```
aggregate(exp_2_data$rt, list(exp_2_data$condition), FUN = mean)
aggregate(exp_2_data$rt, list(exp_2_data$condition), FUN = sd)
```

### Perform paired t-tests

```
```{r Experiment 2 RT test, warning=FALSE, message=FALSE}
t.test(rt ~ condition, data = exp_2_data, paired = TRUE, alternative =
"two.sided")
```
```

```
```{r Experiment 2 recall error test, warning=FALSE, message=FALSE}
```

```
t.test(error ~ condition, data = exp_2_data, paired = TRUE, alternative =  
"two.sided")  
``
```

Dear Reviewers,

Thank you for having taken the time to carefully read and review our manuscript. We appreciate your overall positive remarks and are thankful for each of your valuable comments and suggestions that we have embraced to improve our manuscript.

Prompted by overlapping comments from Reviewers 1 and 3, our most important revision regards that we have now added a critical clarification to our manuscript that carefully delineates what we *can* and what we *cannot* conclude from our data. In short, we now make clear how our data provide proof-of-principle evidence that there is not an automatic or *obligatory* trade-off in the use of space for working memory, when we additionally have time available for item individuation. At the same time, we agree with the reviewers that this does not imply that there may never be such a trade-off – a key clarification that we now discuss extensively and that has led us to also update relevant phrasings throughout our article. For your convenience, we have highlighted each change made in our revised manuscript.

Please find our detailed point-by-point responses to the above as well as each of the other valuable comments and suggestions below. Thank you in advance for reconsidering our manuscript.

Sincerely,

Eelke de Vries, George Fejer, Freek van Ede

Reviewer #1 (expertise: working memory)

This manuscript describes two behavioral studies on spatial reliance in visual working memory. Previous studies have shown that even when location is task-irrelevant, people use space to individuate items in WM. The current study asked if this reliance is preserved or reduced if the WM items can also be individuated by temporal cues. In the baseline Simultaneous task, two WM items (colored tilted bars) were presented simultaneously, then during the WM delay, a central cue indicated the color of the to-be-probed item; after an additional delay, subjects reported the orientation of that remembered item. The key measure is eye position gaze bias during the second delay: the authors replicate prior studies showing that gaze position is biased toward the location of the to-be-probed item, in the absence of visual input. In Experiment 1, the authors compare this “towardness” bias to a Sequential condition, where the 2 WM items are presented sequentially, at different points in time; they report similar gaze bias for sequential and simultaneous conditions. In Experiment 2, all items are presented sequentially, and the to-be-probed item is cued either by its color or its temporal position; again, the authors report similar gaze bias for these conditions. The conclusion is that location reliance is automatic and there is no tradeoff when temporal cues are available.

I thought this was an interesting paper asking a neat and novel question. I liked the studies, though I’m left feeling not quite convinced about the conclusion. To be clear, I think the results definitively show a strong reliance on location information in these scenarios where location is increasingly not needed, and I think that could make a nice contribution to the literature. I’m just less convinced that the “no trade-off” claim is being compellingly tested.

Thank you for your overall positive remarks as well as for each of the constructive comments and suggestions that we respond to below.

In short, as we now make explicit in our revised manuscript, our data primarily serve to provide *proof-of-principle* that there does not *need* to be a trade-off between the use of space and time for visual working memory. We now explicitly acknowledge that our data do not imply that there will never be such a trade-off. Rather, we show that relying on time (most clearly established in Experiment 2 where participants *had* to use temporal order to arrive at the appropriate memory content) does not inevitably come as a cost to using space as well, as we found that the use of space – as reflected in the gaze biases – did not diminish. In addition, please also note how we now provide statistical evidence from Bayesian analysis in favor of no difference between our experimental conditions (see our response to the Editor comment above).

Comment 1. Methodological Concerns

1a. The study aims to test a tradeoff between the use of space and time. But the way the cues are instantiated in the current study doesn’t feel like they are equally strong, from a methodological perspective. For example, space here is always a coarse hemifield distinction. The items are always presented on opposite sides of the hemifield. This is a very obvious spatial cue, with the items encoded by different hemispheres of visual cortex, and that could contribute to the robust/automatic nature of this effect. On the other hand, the temporal cues are of weaker and more relative magnitude (across the entire 1-hr experiment, the temporal differences between the two items on a given trial only differ by 1 second). The spatial tags are 2 consistent, absolute locations across the entire experiment, while the temporal tags are changing across the whole experiment, which unfolds in time. I think this needs discussing, and serious consideration of whether the paradigm allows for a fair testing of a potential “tradeoff”.

Thank you for raising this relevant point. The reviewer is right that it is not trivial to “equate” the spatial and temporal dimensions, and that space may have been more pervasive in our experiment.

Even so, it is clear from Experiment 2 that participants were able to use temporal order information to select the appropriate memory content (if not, participants' performance would be much worse in the order-blocks in Experiment 2, where the relevant memory content could exclusively be selected based on its temporal position).

What our findings reveal is that using temporal order (as was the case most clearly in temporal-order cueing blocks in Experiment 2) does not necessarily reduce the reliance on space. It is in this sense that our manuscript provide “proof-of-principle” that the use of temporal order does not lead to an obligatory trade off to the use of space. This, however, does not imply that there will never be a trade-off. Indeed, if space was less obvious as a feature for item individuation, then it may well be that space would also be less used, especially when other sources of item individuation were available.

Also prompted by overlapping comments from Reviewer 3, we now added this crucial clarification to our Discussion:

The following paragraph has been added to the Discussion section (lines 301:318):
“By showing preserved incidental use of space for mnemonic selection when temporal order information was additionally available (Experiment 1) or necessary (Experiment 2) for selection, our data provide proof-of-principle that the use of time for mnemonic selection does not obligatorily come at a cost to the reliance on space. In other words, we show that a trade-off between the use of space and time is not inevitable. At the same time, we do not wish to claim that such a potential trade-off will never occur – we merely show that it does not necessarily occur. In our experiments, memory objects were consistently presented in distinct hemifields, with clear spatial separation, making space a viable and powerful means to individuate the objects, even when subsequently cued through temporal order. It is conceivable that the use of space may be less prominent when the two objects were to be presented within the same hemifield or closer in proximity, or even temporally separated at an identical spatial location. One potential factor influencing the utilization of space or time for object individuation could be the degree of separability between the objects in either dimension, given that decreased separability may lead to heightened inter-item interference. As such, as spatial separation decreases, the cognitive system may be more inclined to use alternate individuating dimensions, such as time, to minimize interference (Oberauer & Lin, 2017; Sapkota et al., 2016; Schneegans et al., 2021). Exploring the incidental uses of space and time — and potential trade-offs between them — by systematically varying the separability of the objects will thus be an interesting avenue for subsequent research. Moreover, in future work, neuroimaging can be used, in addition to eyetracking, to provide relevant complementary findings about the neural bases of such potential trade-offs (for a recent example, see Fulvio et al., 2023).

In line with this key revision, we have also updated key phrasings in our Abstract and Introduction, such as:

Abstract (lines 18:20): *“[...] Thus, space remains a profound organizing medium for working memory even when other organizing sources are available and utilised, with no evidence for an obligatory trade-off between the use of space and time.”*

Significance statement (lines 28:30): *“[...] This shows there is no obligatory trade-off between spatial and temporal codes available for memory organisation, advancing our understanding of the spatial-temporal architecture of mind.”*

Introduction section (lines 120:121): “[...]. In other words, we report no evidence for an obligatory trade-off in the incidental use of space and time for visual working memory.”

Discussion section (lines 282:285): “[...]. Our results suggest that space remains a profound organizing principle that serves the individuation and selection of visual objects from working memory even when time is also available, or even has to be used – with no evidence for an obligatory trade-off between the incidental use of space and time. ”

Finally, in light of these revisions, we have also considered to update our title to “No **obligatory** trade-off between the use of space and time for working memory” or “Time does not necessarily overrule space: temporal individuation does not undermine spatial indexing in visual working memory”. At the same time, in order to maintain consistency with the *bioRxiv* preprint, and to keep the title simple and short, we have for now decided to leave our title as is, while nuancing our claims in the manuscript where we have more words to do so. However, should the editor and/or reviewers consider this a necessary change, we are willing to update the title too.

1b. Another imbalance is that in Experiment 1, spatial cues are always present and available to individuate, while temporal cues are only available on half of the trials. It feels like a better test would be to have some spatial-only trials (simultaneous), some temporal-only trials (sequential at same location), and some both-available trials (sequential at different locations). The key comparison would still be gaze bias in spatial-only vs both-available, but in this context, space can’t always be used, so it seems like a fairer test of the tradeoff.

We appreciate the suggestion for also including a condition where the two items are presented at the same location. In fact, we initially piloted such a condition. However, during piloting, due to the imbalance in such displays, we observed a gaze bias already prior to the retrocue when both items were presented to the same side of the central fixation cross. This preemptive bias would compromise our primary post-retrocue gaze-bias measure and thereby complicate the comparison to the other two spatially balanced conditions in the post-retrocue phase. It was for this reason that we decided not to also include this condition.

Having clarified this, we nevertheless agree that Experiment 1 did not provide sufficiently conclusive evidence for our main claim that the use of time does not necessarily come as a cost to the use of space (the “proof-of-principle” argument above). It was for this reason that we included Experiment 2 where items were always presented sequentially, and where we instead manipulated what cue feature should be used for item selection whereby in half the blocks participants were instructed to use the temporal order information in the cue while in the other half the blocks they were instructed to use cue colour (building on Experiment 1).

Finally, please note in our response to the comment above, how we have now added a critical clarifying paragraph to our discussion where we explicitly raise the possibility that our observations may be contingent on the specific spatial configuration tested in the experiment. Here we also explicitly suggest that testing a condition with smaller (or even no) spatial separability would be interesting to assess the generalizability of our findings, beyond the proof-of-principle demonstration that a trade-off is not obligatory.

Comment 2. Testing Automaticity of Spatial Indexing

The hypothetical experiment described above would also allow for another stronger test of the automaticity of spatial indexing: Would gaze still be biased toward the item’s location if location

could NOT be used to differentiate? Imagine the condition where the two items are presented sequentially, both on the left side. Then one of the items is cued by either color or temporal position. Would gaze still be biased toward the left side? If so, that supports an automatic, sensory recruitment explanation, regardless of whether space can be used for individuation.

Please see our response above where we outline that we had, in fact, initially included such a condition when piloting this experiment, and why we had eventually decided (based on pilot data) not to include it as a third condition in the experiment.

In addition, as also stated above, we now suggest this as an interesting avenue for future research, even though it might be challenging to directly compare the outcomes from such a spatially unbalanced condition to the spatially balanced conditions that we ended up including and comparing.

Comment 3. Interpretation experiment 2

I appreciate the motivation for Experiment 2, having a condition that explicitly uses temporal order to cue the to-be-reported item. But the experiment doesn't seem to be directly testing a space-time tradeoff, more of a color-time tradeoff. This is not to say I don't find the existence of the spatial bias in the order condition compelling. But using it as evidence for "no tradeoff between the use of space and time" (title) feels like too much of an over-reach.

We appreciate your point on the interpretation of Experiment 2. Our objective was to examine whether the incidental use of space as a scaffold for visual working memory would be compromised when time had to be used for item individuation/selection.

As an alternative to colour cues, we could in principle have used spatial cues. However, in such a case, we would no longer be investigating the *incidental*, scaffolding use of space. Moreover, spatial cues could have themselves triggered gaze biases that reflected bottom-up driven cue processing, rather than top-down cue-triggered mnemonic selection through memorised location.

Though we used colour cues, please note how our measure of interest – directional biases in gaze – informed about the use of space for working memory, thus allowing us to address potential trade-offs in the use of space. In several prior studies, we have consistently shown how colour cues can provide an excellent vehicle for triggering space-based selection of mnemonic content. In Experiment 2 we therefore again used colour cues as our baseline (also to build directly on Experiment 1), and asked whether the incidental use of space would be reduced when participants were instead forced to rely on the temporal-order feature of the cue instead. The lack of this hypothetical reduction in the use of space (as indexed through gaze) in turn suggests that the use of temporal order does not necessarily come at a cost (or trade-off) to the incidental use of space for mnemonic selection. We now explicitly clarify our rationale for again relying on colour cues in Experiment 2 in our Methods section:

The following text has been added Methods section (lines 435:444): “[...]. *In Experiment 2, we again relied on colour cues for several reasons. First, the use of colour cues ensured methodological consistency with Experiment 1, enabling us to build cohesively upon its findings. Second, our previous research has consistently highlighted the effectiveness of colour cues in evoking space-based mnemonic content selection (Chawoush et al., 2023; Liu et al., 2022; van Ede et al., 2019, 2020, 2021; de Vries & van Ede, 2023). While we also considered directly using spatial cues, this would have had several drawbacks. First, introducing spatial cues would shift the focus away from incidental spatial engagement, which is the study's underlying foundation, and toward direct spatial referencing. Second, when using*

spatial cues, the gaze biases that were studied here could be directly driven by cue processing rather than reflecting mnemonic space-based selection. Color cues have the advantage of not causing any spatial biases in gaze directly due to the cue's bottom-up stimulus features. [...]”

Comment 4. Statistical Analysis

The statistical tests were focused on when each condition individually was significant from zero. But unless I missed it, I didn't see any direct comparisons between conditions. While it's clear there is a strong bias in all conditions, were there any differences in the towardness metric, e.g. in terms of amplitude or duration? In Experiment 1, it looks like there's a consistently stronger and more prolonged bias in the simultaneous condition, and in Experiment 2, the bias in the color condition appears to peak earlier and higher.

Thank you for this valuable feedback. We had initially compared conditions using cluster-based permutation analysis and found no significant clusters in the relevant comparisons (despite very robust clusters for the biases in each of the conditions by themselves, as correctly pointed out by the reviewer). We apologize that we initially forgot to mention this lack of significant clusters in the relevant condition comparisons. We have now corrected this. In addition, prompted by this comment, as well as by the Editorial recommendation, we performed additional Bayesian t-tests, directly comparing conditions for towardness and saccade-rates time-courses. These analyses provided additional support for our original conclusions and have been added to our manuscript, as stated above in our response above to the editorial comments.

Reviewer #2 (expertise: eye-tracking)

This is a nice paper in which two experiments aimed at revealing the possible relationship between space and time in WM are reported. Both experiments relied on a simple task in which the position or the colour of the stimulus had to be memorised while eye movements were recorded. In summary, space appears to be an important dimension for WM even when other information could be used to complete a task. I have found this work to be well-organised and written and I have little to say.

Thank you for your overall positive remarks as well as for the constructive comments and suggestions that we respond to below.

Comment 1. Sample Size Justification

My main comment is related to sample size: I think a subjective criterion (e.g., $N = 25$ because other studies have done something similar) could open several questions and a power analysis is always preferable. So, I am wondering if more details can be added to provide a more solid justification of the priori-established sample size used in both experiments.

We appreciate the reviewer's suggestion regarding sample-size justification. Our decision to include 25 participants in both experiments was initially based on several considerations.

First, given that our main outcome measures involved time courses that consist of “effects” at many time points, it is not trivial to determine the power, unless one averages these time courses over a specific time range (and it is not immediately trivial to determine what time range this should be to estimate the power). Instead of averaging these time courses over some arbitrary time range, we based our sample size on several previous studies using the same outcome measure. These studies consistently yielded robust findings with this sample size (e.g., Liu et al., 2022; van Ede et al., 2019, 2020, 2021).

Second, by maintaining the same a-priori determined sample size as used in previous work, for both of our experiments, we aimed to ensure consistency and comparability between the studies. This approach allowed us to effectively replicate the key outcome measure and assess its stability across independent experiments, including here between Experiments 1 and 2.

Nevertheless, we understand and appreciate the reviewer's comment regarding our sample size determination. We agree that an objective, quantitative justification could strengthen the confidence in our study. To this end, we now conducted a sensitivity analysis using the 'pwr' package in R (Champely, 2020). This analysis is a recognized statistical method that calculates the minimum effect size our study is powered to detect, given the chosen sample size, significance level, and desired power.

The following text has been added to the Methods section (lines 374:378): "[...]. Through a sensitivity analysis conducted with the 'pwr' package in R (Champely, 2020), we confirmed that our within-subject design, involving 25 participants in each of two conditions per experiment and a significance threshold of $\alpha = 0.05$, provides 80% power to detect an effect size of medium to large magnitude (Cohen's $d = 0.584$). This indicates that our study was sufficiently powered to detect any meaningful effect of a reasonable magnitude, although smaller effects might have remained undetected. [...]"

Comment 2. Literature Coverage

My second comment is about the coverage of the literature. In recent years microsaccades have been ‘rediscovered’ and in particular their relationship with higher cognitive mechanisms. In this regard, I am missing some studies reporting a link between these tiny eye movements and working memory, operationalised at different levels and with different tasks, such as mental arithmetic (Siegenthaler et al., 2014) or counting (Valsecchi et al., 2007), or with Sternberg’s task (Dalmaso et al., 2017). I think it would be fair for these studies to be added and briefly commented on in the general discussion.

Thank you for pointing us to this interesting and relevant work. We appreciate the reviewer's insightful comment regarding the relationship between microsaccades and working memory. In light of this comment, we now also highlight these relevant studies investigating the link between microsaccades and working memory.

The following text has been added to the Discussion section (lines 335:340):
“Recent studies have uncovered the role of microsaccades in various cognitive functions, including their decreased rates with increased memory loads (Dalmaso et al., 2017; Kadosh et al., 2023), prolonged inhibition following the presentation of task-relevant oddball stimuli (Valsecchi et al., 2007), and variations in rates and magnitudes with mental arithmetic task difficulty (Siegenthaler et al., 2014; Gao et al., 2015). Complementing this work on microsaccade rates, our study specifically focused on spatial biases in the direction of microsaccades (Engbert & Kliegl, 2003; Hafed & Clark, 2002; Yuval-Greenberg et al., 2014; Corneil & Munoz, 2014), building on other recent studies demonstrating that such spatial biases are also observed when shifting attention among the contents of visual working memory (Draschkow et al., 2022; Liu et al., 2022; de Vries & van Ede, 2023; van Ede et al., 2019, 2020, 2021). [...].”

Reviewer #3 (expertise: visual working memory)

This manuscript examines the role of space and time in visual working memory – specifically whether observers will alleviate their use of space when separation in time allows object individuation. In the first experiment, two colored bars with varying orientations are presented simultaneously (two at a time; sometimes early, late or both early and late timepoints) or sequentially (one at a time; left then right, or vice versa). Participants were cued to an item with a color and required to report the orientation of that item. In the second experiment, the items were always presented sequentially, but the retro-cue was either a color or the temporal order (first or second). The authors track microsaccades towards a cued item as an operationalization for use of space (spatial gaze bias) and find that spatial gaze biases remain unchanged despite the inclusion of temporal individuation, suggesting that there is no trade-off of time and space, and that space is a strong organizational property for working memories.

I think the issue at hand is very current and critical to the study of visual working memory – the empirical results are likely of interest to the wider field. However, I am unconvinced that the manipulations employed in the presented experiments ‘forced’ participants to retrieve via a temporal index in a way that would be at the cost of the spatial index. That is, I’m not sure how far the empirical finding will generalize. I hope the authors find the following comments constructive in improving their manuscript.

Thank you for your overall positive remarks as well as for each of the constructive comments and suggestions that we respond to below. Please note how several responses below overlap with our responses to conceptually similar comments from Reviewer 1.

Comment 1. Suggestions repository

I commend the authors for making their data and code openly accessible on the Open Science Framework – I accessed these files to inform my review. I also confirmed using statcheck.io that the t-statistics and p-values reported in the manuscript are consistent.

- a) The repository could use some README files to explain the variables in the raw data .txt files, as well as a brief explanation of the different functions attached. This will increase the reproducibility of the studies presented in the manuscript.
- b) Could the authors also upload the ‘beh_subject_avg.csv’ file or the function for the iterative exclusion procedure of slow response times?

I tried to reproduce the behavioral analyses (paired t-tests on response onset times and recall accuracy) using the ‘behavior_all_subjects.csv’ in the ‘processed_data’ folder, before realizing that the processed data file likely contains all trials before exclusion due to slow response onset times.

It is great to learn that the reviewer has used our openly shared data to inform the review, and we thank the reviewer for the valuable user feedback that we have embraced to improve the quality of our data sharing. We have now included comprehensive README files that explain the variables in the raw data .txt files and that provide a brief explanation of the different scripts and functions used in the study. Furthermore, in line with enhancing the clarity and comprehensibility of the codebase, we have also embedded documentation at the beginning of all the scripts and functions used in our repository. This includes detailed explanations of the purpose, inputs, outputs, and workings of each piece of code. This effort ensures that other researchers can easily understand and potentially utilize our code in their work.

Additionally, we now uploaded files with the trial-by-trial task performance including which trials were excluded (behaviour_all_subjects_rejectVariable.csv) and the relevant subject averages after

excluding these trials (beh_subject_avg.csv). The iterative exclusion procedure is described in the manuscript (Section: Data acquisition & preprocessing) and was implemented in Matlab via the 'b_preprocess2.m' script (which outputs beh_subject_avg.csv).

Comment 2. Re-Analysis Findings

Doing this re-analysis revealed that most reported effects were robust to data cleaning procedures – the one result differing to what was reported in the manuscript was the response onset times for simultaneous versus sequential presentations before trial exclusions (significantly slower for simultaneous than sequential). This suggests that the simultaneous condition elicited more extremely delayed responses – perhaps without the temporal index available, retrieval of the feature value via the spatial index is slower, but with it available, retrieval is speeded and lowers response onset times. The proportion of excluded trials should be reported by condition within each experiment, rather than by experiment (page 9, line 420). I have attached my reanalysis as a .Rmd file and .html file for reference.

We apologize for the initial oversight in reporting the proportion of excluded trials per condition. In the revised manuscript, we now provide a detailed breakdown of the excluded trials per condition within each experiment, following the valuable and rightful suggestion by the reviewer.

The following paragraph has been added to the Methods section (lines 462:475):
"[...] To enhance data sensitivity, we employed two specific exclusion criteria. First, any trial in which the gaze position surpassed 50 normalized units was omitted, identical to the procedure used in (van Ede et al., 2020). Second, we used participants' response-onset times to estimate their attentiveness to the task and removed trials that had a response onset slower than the mean response onset + 4 standard deviations (following an iterative procedure until no more RT-outliers would be left). For Experiment 1, an average of 2.90% of trials (SD = 5.09%) were excluded. Breaking down by specific criteria, due to excessive eye movements, 1.20% (SD = 4.77%) of trials from the sequential condition and 1.39% (SD = 5.20%) from the simultaneous condition had to be removed. Due to inattentiveness, 1.38% (SD = 1.11%) of trials from the sequential condition and 1.92% (SD = 0.85%) from the simultaneous condition were removed. For Experiment 2, an average of 2.81% of trials (SD = 2.69%) were excluded. Specifically, due to excessive eye movements, 1.20% (SD = 2.26%) of trials from the colour condition and 1.45% (SD = 2.93%) from the order condition were removed. As for inattentiveness, 1.44% (SD = 0.90%) of trials from the colour condition and 1.60% (SD = 0.95%) from the order condition were excluded. Taken together, these results confirm that our task did not induce substantial eye movements following the central retrocue."

We also appreciate the reviewer's thorough analysis and observation regarding the longer RTs in the simultaneous condition prior to outlier exclusion. While it's a noteworthy finding, we wish to refrain from overinterpreting it, given we had defined our outlier criteria a-priori. Moreover, our primary focus was directed towards the gaze data, while the behavioural performance data mostly served to verify participants were able to do the task and that participants used the cues appropriately (in Experiment 2).

It is further useful to acknowledge that, occasionally, participants took breaks or there were instances where the eye-tracker momentarily lost the participant's gaze due to head movements. During these minor hiccups, participants were prompted to delay their responses. This could have led to some of the extreme outliers in the data. As a result, these prolonged response times might not

solely be attributable to the cognitive reasons posited but could also be influenced by these external factors.

That said, we agree that the breakdown of removed trials per condition is an important improvement that increases the transparency of our report.

Comment 3. Critique of Interpretation

The authors suggest that because there was no difference in spatial gaze bias when retro-cued with a temporal index compared to with the color of the item (i.e. spatial organization is still used with temporal indexing) that “the use of space remains important, even when time can, or even has to, additionally be used for object individuation in working memory” (page 7, line 331). I don’t think this has been definitively shown yet – in my view, the experimental conditions do not impose a significant enough load where spatial organization may be lost. It is interesting that participants retain spatial organization of the items when retro-cued with a temporal index, but this may be because that spatial organization helps boost retrieval of the featural value and there is no impetus to drop the spatial organization. One might expect that when the discriminability of spatial index is difficult (objects are presented close together) that the use of spatial information may be less relied upon (and the temporal index may be more relied upon, though to my knowledge we don’t have a measure for this reliance).

A potential follow-up experiment that might be more convincing would be to include trials where the spatial organization hinders the retrieval of the featural value – say, if items are presented on the same side (similar spatial locations) forcing subjects to better individuate through temporal order. Participants’ spatial gaze bias may be suppressed as a result of the spatial index being confusing. This new condition would likely need to include some spatial placeholders to keep the visual center of mass around the fixation cross, as was done in the experiments presented so far. Comparing the degree of spatial bias in those conditions compared to when the items are presented at clear separate spatial locations (left and right) as was done here would be a more definitive test. Observing a spatial gaze bias that is no different would then be a more convincing demonstration that there is no trade-off.

Thank you for raising this relevant point. In response to a similar comment from Reviewer 1, we now explicitly and carefully state which claims we *can* and *cannot* make based on our data.

What our data show is that when temporal order information must be used for item selection (Experiment 2, cue-order blocks) that this does not inevitably cause a reduction in the incidental use of location for working memory. As such we provide “proof-of-principle” that the use of time does not automatically or obligatorily reduce (trade-off) the use of space.

At the same time, we cannot (and do not wish to) claim that there will *never* be a trade-off between the use of space and time – all we claim is that there does not *have* to be one. We agree with the reviewer that if one would make the use space less potent (e.g., because items would be in the same hemifield, close to each other), that it is conceivable that space may become less important when other sources of item individuation are available, such as temporal order or another feature.

Prompted by a similar comment from Reviewer 1, we have now added the following critical clarification on this point to our Discussion:

The following paragraph has been added to the Discussion section (lines 301:318):
“By showing preserved incidental use of space for mnemonic selection when temporal order information was additionally available (Experiment 1) or necessary (Experiment 2) for selection, our data provide proof-of-principle that the use of time

for mnemonic selection does not obligatorily come at a cost to the reliance on space. In other words, we show that a trade-off between the use of space and time is not inevitable. At the same time, we do not wish to claim that such a potential trade-off will never occur – we merely show that it does not necessarily occur. In our experiments, memory objects were consistently presented in distinct hemifields, with clear spatial separation, making space a viable and powerful means to individuate the objects, even when subsequently cued through temporal order. It is conceivable that the use of space may be less prominent when the two objects were to be presented within the same hemifield or closer in proximity, or even temporally separated at an identical spatial location. One potential factor influencing the utilization of space or time for object individuation could be the degree of separability between the objects in either dimension, given that decreased separability may lead to heightened inter-item interference. As such, as spatial separation decreases, the cognitive system may be more inclined to use alternate individuating dimensions, such as time, to minimize interference (Oberauer & Lin, 2017; Sapkota et al., 2016; Schneegans et al., 2021). Exploring the incidental uses of space and time — and potential trade-offs between them — by systematically varying the separability of the objects will thus be an interesting avenue for subsequent research. Moreover, in future work, neuroimaging can be used, in addition to eyetracking, to provide relevant complementary findings about the neural bases of such trade-offs (for a recent example, see Postle et al., 2023).

In line with this key revision, we have also updated key phrasings in our Abstract and Introduction, such as:

Abstract (lines 18:20): *"[...]. Thus, space remains a profound organizing medium for working memory even when other organizing sources are available and utilised, with no evidence for an obligatory trade-off between the use of space and time."*

Significance statement (lines 28:30): *"[...]. This shows there is no obligatory trade-off between spatial and temporal codes available for memory organisation, advancing our understanding of the spatial-temporal architecture of mind."*

Introduction section (lines 120:121): *"[...]. In other words, we report no evidence for an obligatory trade-off in the incidental use of space and time for visual working memory."*

Discussion section (lines 282:285): *"[...]. Our results suggest that space remains a profound organizing principle that serves the individuation and selection of visual objects from working memory even when time is also available, or even has to be used – with no evidence for an obligatory trade-off between the incidental use of space and time."*

Finally, in light of these revisions, we have also considered to update our title to *"No **obligatory** trade-off between the use of space and time for working memory"* or *"Time does not necessarily overrule space: temporal individuation does not undermine spatial indexing in visual working memory"*. At the same time, in order to maintain consistency with the *bioRxiv* preprint, and to keep the title simple and short, we have for now decided to leave our title as is, while nuancing our claims in the manuscript where we have more words to do so. However, should the editor and/or reviewers consider this a necessary change, we are willing to update the title too.

While we appreciate the idea for potential further experiments in which the spatial positioning of the items is systematically varied, it is useful to highlight the challenge this would present. Specifically, presenting the two items at the same distal location leads to a pre-retrocue gaze bias towards the side of both items. This was evident from our pilot studies, where we initially included such a condition (also mentioned in our response to a similar comment from Reviewer 1). Such a pre-retrocue bias makes it difficult to compare conditions, especially when comparing to the two conditions that we did include, where the items were always spatially balanced around the central fixation cross (as foreseen by the reviewer). It was for this reason that we decided not to also include this condition. At this point, instead of running additional experiments, we decided to be more explicit about what our data allows us to claim (that it is not *obligatory* to have a trade-off) and what they do not allow us to claim (that there will *never* be a trade-off). Furthermore, as detailed in the added paragraph above, we now encourage future research to systematically vary both spatial and temporal separability of the items to test if and when such a potential trade-off might arise. Even so, we believe our data provide an important advance by showing that the use of space can remain fully preserved, even when other sources (i.e. time) must be used for item individuation.

Comment 4. Interference model

A useful framework for the present set of studies is one proposed by the Interference Model (Oberauer and Lin, 2017). They propose that similarity in the context (the feature dimension cued for retrieval) will produce more interference, resulting in more errors in reporting the correct feature value. It is likely that the observer may trade-off between the spatial and temporal conditions when discriminability in one of those dimensions is low and the other is high. Here, the present studies only test cases when spatial discriminability is high, perhaps masking any effect of trade-off. Note that Schneegans, McMaster and Bays (2022) in similar experiments find both location and ordinal (temporal) position mediate binding and have similar contributions to error correlations.

Thank you for bringing our attention to the framework suggested by Oberauer and Lin (2017), as well as the relevant findings by Schneegans et al. (2022). As we address in the newly added paragraph (see our response to the above comment), our main contribution is to show that there does not *need* to be a trade-off between space and time. In other words, space can remain used to a similar extent even when items are cued through temporal order. Still, we agree that in future studies it would be interesting to systematically manipulate spatial and temporal discriminability to assess whether this may modulate and/or generate trade-offs in the use of specific dimensions, such as space and time. Prompted by the reviewer's comment, we have incorporated this suggestion, along with the studies highlighted by the reviewer, into the new paragraph above.

27th Oct 23

Dear Mr de Vries,

Your manuscript titled "No trade-off between the use of space and time for working memory" has now been seen by our reviewers, whose comments appear below. In light of their advice I am delighted to say that we are happy, in principle, to publish a suitably revised version in Communications Psychology under the open access CC BY license (Creative Commons Attribution v4.0 International License).

We therefore invite you to revise your paper one last time to address the remaining concerns of our reviewers and a list of editorial requests. At the same time we ask that you edit your manuscript to comply with our format requirements and to maximise the accessibility and therefore the impact of your work.

Please note that it may still be possible for your paper to be published before the end of 2023, but in order to do this we will need you to address these points as quickly as possible so that we can move forward with your paper.

EDITORIAL REQUESTS:

As you will see, Reviewer #1 highlights persistent concerns about the lack of a control experiment. Although we editorially decided upon consultation that the advance offered by the present study was of sufficiently broad interest to our readership in the absence of the additional experiments, we do expect the referees' feedback to be reflected in the Discussion, which presently lacks the mandatory "Limitations" section.

SUBMISSION INFORMATION:

OPEN ACCESS:

Communications Psychology is a fully open access journal. Articles are made freely accessible on publication under a [CC BY](http://creativecommons.org/licenses/by/4.0) license (Creative Commons Attribution 4.0 International License). This license allows maximum dissemination and re-use of open access materials and is preferred by many research funding

bodies.

For further information about article processing charges, open access funding, and advice and support from Nature Research, please visit <https://www.nature.com/commspsychol/article-processing-charges>

At acceptance, you will be provided with instructions for completing this CC BY license on behalf of all authors. This grants us the necessary permissions to publish your paper. Additionally, you will be asked to declare that all required third party permissions have been obtained, and to provide billing information in order to pay the article-processing charge (APC).

* **DATA AVAILABILITY:**

[link redacted]

Best regards,

Jesse Rissman &
Antonia Eisenkoeck

Antonia Eisenkoeck
Senior Editor
Communications Psychology

REVIEWERS' COMMENTS:

Reviewer #1 (Remarks to the Author):

The authors did a nice job revising the manuscript to more accurately convey what can and cannot be concluded from this study. I do think the title should also be modified to reflect this; bioRxiv preprints can still be linked across versions with different titles, so this shouldn't be a reason to stick with a less appropriate title. Overall, I continue to find this a good, interesting paper that will make a nice contribution to the literature, but I am not sure that with these weakened conclusions (and the decision to not pursue additional experiments that could have strengthened them), that the theoretical impact is high enough for this journal.

Reviewer #2 (Remarks to the Author):

I am happy with the revised version of this paper.

Reviewer #3 (Remarks to the Author):

The authors have made diligent changes to the manuscript in response to the reviewers' comments. Of note, the authors have clarified their conclusions on the roles of space and time in working memory – that there is no obligatory trade-off between space and time. The authors have also updated the associated data and code repositories and ensured its computational reproducibility. I believe the work has shown the value of studying microsaccades to understanding visual working memory, and opens the doors to better understanding of the contributions of space and time in working memory encoding and representaiton.

1. I commend the authors for their efforts in updating their Open Science Framework repository – it is now well organized and substantially improved for computational reproducibility. I exactly reproduced the t-tests in the manuscript (see attached .html file), and I confirmed that the provided `f_plot_figures.html` matches the results presented in the manuscript.

Signed,
William Xiang Quan Ngiam

COMMSPSYCHOL-23-0116A

William Ngiam

2023-10-21

```
all_data <- read.csv("beh_subject_avg.csv")

#Experiment 1
E1_acc_data <- all_data %>%
  filter(experiment == 1) %>%
  select(subject,condition,accuracy_mean) %>%
  pivot_wider(id_cols = subject, names_from = condition, values_from = accuracy_mean)

t.test(E1_acc_data$Sequential,E1_acc_data$Simultaneous, paired = TRUE)
```

Paired t-test

```
data: E1_acc_data$Sequential and E1_acc_data$Simultaneous
t = 5.8025, df = 24, p-value = 5.549e-06
alternative hypothesis: true mean difference is not equal to 0
95 percent confidence interval:
 0.9424907 1.9831031
sample estimates:
mean difference
 1.462797
```

```
E1_rt_data <- all_data %>%
  filter(experiment == 1) %>%
  select(subject,condition,rt_mean) %>%
  pivot_wider(id_cols = subject, names_from = condition, values_from = rt_mean)

t.test(E1_rt_data$Sequential,E1_rt_data$Simultaneous,paired = TRUE)
```

Paired t-test

```
data: E1_rt_data$Sequential and E1_rt_data$Simultaneous
t = -1.6751, df = 24, p-value = 0.1069
alternative hypothesis: true mean difference is not equal to 0
95 percent confidence interval:
-31.536379  3.278961
sample estimates:
mean difference
-14.12871
```

```
#Experiment 2
E2_acc_data <- all_data %>%
  filter(experiment == 2) %>%
  select(subject,condition,accuracy_mean) %>%
  pivot_wider(id_cols = subject, names_from = condition, values_from = accuracy_mean)

t.test(E2_acc_data$Color,E2_acc_data$Order,paired = TRUE)
```

Paired t-test

```
data: E2_acc_data$Color and E2_acc_data$Order
t = -3.6642, df = 24, p-value = 0.001225
alternative hypothesis: true mean difference is not equal to 0
95 percent confidence interval:
 -2.0265235 -0.5661707
sample estimates:
mean difference
 -1.296347
```

```
E2_rt_data <- all_data %>%
  filter(experiment == 2) %>%
  select(subject,condition,rt_mean) %>%
  pivot_wider(id_cols = subject, names_from = condition, values_from = rt_mean)

t.test(E2_rt_data$Color,E2_rt_data$Order,paired = TRUE)
```

Paired t-test

```
data: E2_rt_data$Color and E2_rt_data$Order
t = 0.21912, df = 24, p-value = 0.8284
alternative hypothesis: true mean difference is not equal to 0
95 percent confidence interval:
 -9.580571 11.856534
sample estimates:
mean difference
 1.137982
```